# More Than Meets the Eye? Uncovering the Reasoning-Planning Disconnect in Training Vision-Language Driving Models

## Abstract

Vision-Language Model (VLM) driving agents promise explainable end-to-end autonomy by first producing natural-language reasoning and then predicting trajectory planning. However, whether planning is **causally** driven by this reasoning remains a critical but unverified assumption. To investigate this, we build Drive-Mind, a large-scale driving Visual Question Answering (VQA) corpus with plan-aligned Chain-of-Thought (CoT), automatically generated from nuPlan. Our data generation process converts sensors and annotations into structured inputs and, crucially, separates priors from to-be-reasoned signals, enabling clean information ablations. Using DriveMind, we train representative VLM agents with Supervised Fine-Tuning (SFT) and Group Relative Policy Optimization (GRPO) and evaluate them with nuPlan's metrics. Our results, unfortunately, indicate a consistent **causal disconnect** in reasoning-planning: removing ego/navigation priors causes large drops in planning scores, whereas removing CoT produces only minor changes. Attention analysis further shows that planning primarily focuses on priors rather than the CoT. Based on this evidence, we propose the Reasoning-Planning Decoupling Hypothesis, positing that the training-yielded reasoning is an ancillary byproduct rather than a causal mediator. To enable efficient diagnosis, we also introduce a novel, training-free probe that measures an agent's reliance on priors by evaluating its planning robustness against minor input perturbations. In summary, we provide the community with a new dataset and a diagnostic tool to evaluate the **causal fidelity** of future models.

## 1 Introduction

End-to-end autonomous driving learns planning directly from sensor data and has attracted sustained attention in both academia and industry commaai (2025); Chen et al. (2024); Hu et al. (2023); Jiang et al. (2023). Recent studies explore Vision Language Model (VLM) driving agents that combine the reasoning capability of large language models (LLMs) with visual perception in order to approximate human driving Wen et al. (2024); Zhang et al. (2024a). Chain of Thought (CoT) Wei et al. (2022) has been shown to enhance reasoning in LLMs Feng et al. (2023), and it is increasingly adopted in VLM driving agents to make the sequence of perception, analysis, and decision explicit Sima et al. (2025); Tian et al. (2024); Wang et al. (2024). The intention is to strengthen planning while improving interpretability and controllability. In this paradigm, the model generates a response that first articulates a CoT for reasoning, followed by the final planning trajectory. Consequently, planning is taken for granted as causally driven through the preceding CoT reasoning.

Despite rapid progress, whether planning in current VLM driving agents is causally mediated by their reasoning remains insufficiently verified. Existing works Xu et al. (2024); Wang et al. (2025) primarily report trajectory quality and rule compliance, which assess how well the planning appears, but not which information pathway produce it. As a result, strong scores cannot be taken as evidence that reasoning contributes causally to planning. Under these conditions, shortcut learning Geirhos et al. (2020); Yuan et al. (2024) is a central risk: a model can obtain high planning scores by exploiting biased or spurious priors, such as ego state and history, rather than by using reasoning to construct the planning. In such cases, the produced reasoning may be only an ancillary byproduct.

To rigorously investigate this causal link, a dataset with plan-aligned CoT is necessary. However, existing datasets fall short of this need. Many real-world datasets Sima et al. (2025); Qian et al.

(2024) use the nuScenes Caesar et al. (2020) benchmark as their foundation. While nuScenes provides high-fidelity sensor data, it inherently lacks the fine-grained semantic annotations, such as traffic light states, speed limits, or complex lane topology, which are essential for deep reasoning. To circumvent the semantic limitations inherent to nuScenes, some researches Wang et al. (2024) have turned to simulation platforms such as CARLA Dosovitskiy et al. (2017). Despite offering high controllability, these platforms face a significant sim-to-real gap Delavari et al. (2025) . Their trajectories are governed by idealized dynamics and often fail to capture the nuanced and at times imperfect behaviors characteristic of real-world human driving.

To bridge this gap, we introduce DriveMind, a novel dataset built upon the nuPlan Karnchanachari et al. (2024) benchmark, specifically curated to facilitate the causal analysis of VLM-based driving agents. We choose nuPlan as our foundation because it uniquely combines the authenticity of large-scale, real-world driving data with the rich, vectorized semantic context necessary for complex reasoning. On this base, DriveMind covers approximately $50,000$ samples spanning 61 driving scenarios, providing broad and diverse coverage for analysis. Another core contribution of DriveMind is the modular organization of the dataset's multi-modal inputs, structuring elements like visual data, ego state, and navigation into distinct modules. This design is critical for conducting controlled ablation studies. By selectively withholding specific information modalities, we can systematically dissect the information flow within a VLM agent and robustly attribute the final planning decisions to either high-level reasoning or low-level shortcut signals.

Using our DriveMind dataset, we train and evaluate representative VLM-based driving agents Bai et al. (2025); Liu et al. (2023b); Wang et al. (2025). We uncover a striking result: an agent trained solely on textual priors, with no visual input and no CoT reasoning, achieves planning scores that match or even exceed those of a fully multimodal counterpart. This reliance on shortcuts is so entrenched that even applying an advanced policy alignment method from reinforcement learning (RL), Group Relative Policy Optimization (GRPO), fails to substantively restore the causal link from reasoning to planning. These findings motivate our Reasoning-Planning Decoupling Hypothesis: the planning module from an agent's output predominantly relies on textual priors (i.e., ego state, history) as shortcuts, largely ignoring the visual context (i.e., surroundings, traffic signals) and the CoT reasoning. To substantiate this hypothesis, we introduce a sequence-level attention analysis, which demonstrates the dominance of prior and ego-state tokens over visual and CoT tokens.

Finally, we propose a generalizable, training-free diagnostic method to distinguish between genuine, CoT-grounded planning and shortcut learning, with the aim of establishing a plug-and-play standard for model evaluation. Grounded in the principles of causal intervention, our method acts as a "causal probe" by applying minor, semantically plausible perturbations to the textual priors (e.g., slight variations in ego states or historical positions). A planner that truly grounds decisions in visual evidence and CoT should be stable under such perturbations, whereas a shortcut-reliant planner will show brittle sensitivity to the exact prior pattern. Therefore, a disproportionately large degradation in planning scores following perturbation indicates a high degree of shortcut learning. Conversely, a robust planning performance signifies that the agent's decisions are properly grounded in a holistic understanding of the driving scene. This method provides a simple yet powerful tool for assessing the causal fidelity of VLM agents. In summary, the main contributions of our paper are as follows:

- We propose and validate the Reasoning-Planning Decoupling Hypothesis, which posits that current training paradigms are insufficient to forge a causal link between reasoning and planning, leading agents to instead learn shortcuts from textual priors.

- We introduce DriveMind, a large-scale nuPlan-based dataset with plan-aligned CoT and a modular design, enabling causal analysis and systematic ablation studies of VLM driving agents.

- We introduce the Causal Probe, a novel, training-free diagnostic method that detects shortcut reliance, providing a new tool for evaluating the causal robustness of driving agents.

## 2 RELATED WORK

**End-to-End Autonomous Driving.** Unlike hierarchical driving models ApolloAuto (2025), end-to-end autonomous driving aims to learn directly from sensor inputs and output planning trajectories, thereby avoiding error accumulation and facilitating optimization. Most existing end-to-end autonomous driving models are based on vector representation learning Chen et al. (2024). Frameworks such as UniAD Hu et al. (2023) and VAD Jiang et al. (2023) transform perception information

into Bird Eye View (BEV) feature vectors for subsequent trajectory planning. ViDAR Yang et al. (2024) and UAD Guo et al. (2024) proposed self-supervised pre-training algorithms, enhancing the scalability of vector representation-based driving models. More recently, the remarkable success of LLM Bubeck et al. (2023); Ahn et al. (2024); Jiang et al. (2025) and their multimodal extension, VLMs Dosovitskiy et al. (2021); Liu et al. (2023b); Zhang et al. (2024b), has introduced a new frontier. Researchers have begun to leverage the powerful feature extraction capabilities of VLMs to enhance the vectorized paradigm. For instance, models like VLP Pan et al. (2024) and DiMA Hegde et al. (2025) distill prior knowledge from pre-trained VLMs into the driving model to improve planning performance. However, both the purely vectorized models and these early VLM-enhanced frameworks fundamentally rely on implicit feature transformations for decision-making. They lack an explicit, human-understandable reasoning process. This opacity not only limits their interpretability but also raises concerns about their robustness.

**Driving Agent with Reasoning.** To enhance the interpretability of VLMs and adapt their capabilities for the driving domain, recent research has focused on optimizing models toward reasoning-based planning. Early approaches often employed multi-turn dialogue paradigms to elicit complex reasoning. For instance, DriveLM Sima et al. (2025) introduced a graph-based VQA dataset that requires the model to first perceive and predict before a human manually selects key context for planning. Similarly, DriveVLM Tian et al. (2024) proposed a hierarchical thinking dataset structured around scenario description, analysis, and planning. However, this multi-turn paradigm incurs significant data annotation and inference overhead, posing challenges to scalability and real-time application. To improve efficiency, subsequent work has shifted towards single-turn dialogue incorporating CoT. DriveCoT Wang et al. (2024), for example, explicitly integrates reasoning and planning into a single model response, but its reliance on the CARLA Dosovitskiy et al. (2017) simulation environment raises key questions about sim-to-real transferability. Omnidrive Wang et al. (2025) trains an agent for implicit reasoning via multi-task VQA. Meanwhile, Omnidrive recognizes the model's dependency on priors like ego state and attempts to mitigate this using counterfactual reasoning. However, as our subsequent experiments will reveal, this method has limited effectiveness in addressing shortcut learning from textual priors. Furthermore, its dataset is built upon nuScenes, inheriting the limitations of semantic richness required for deep causal analysis.

## 3 METHODOLOGY

This section outlines our methodology, which includes four parts. First, we introduce the DriveMind dataset and its generation pipeline. Second, we describe the ablation training of VLM driving agents on DriveMind. Third, we present our sequence-level attention analysis for interpreting information flow. Finally, we introduce our perturbation-based causal probe for assessing shortcut learning.

### 3.1 DRIVEMIND DATASET GENERATION PIPELINE

To investigate the causal relationship between reasoning and planning in VLM-based driving agents, we construct the DriveMind dataset. Existing datasets either lack the complexity of the real world due to being simulation-based or lack the rich semantic information required to support deep rea-

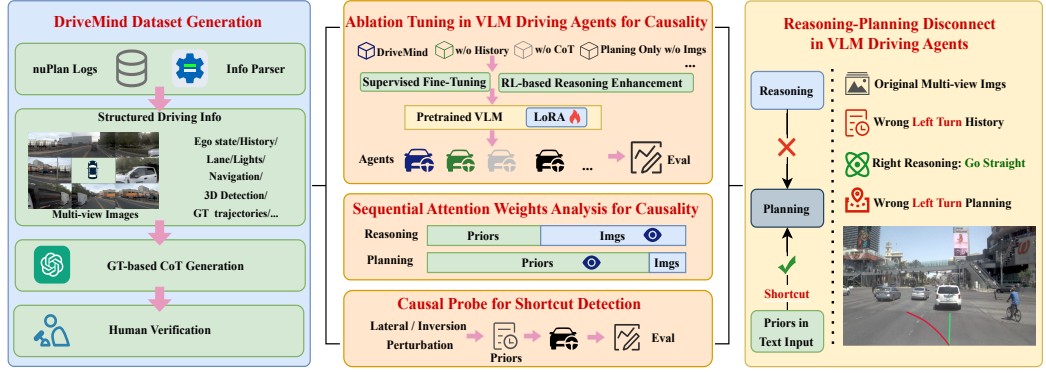

Figure 1: An overview of our framework for investigating the reasoning-planning disconnect in VLM agents. (Left) The DriveMind dataset generation pipeline. (Middle) Our methodology for causal validation. (Right) An illustration of our core finding, where planning follows a textual prior shortcut, bypassing the correct CoT reasoning.

soning. Therefore, the design of DriveMind adheres to four core principles, including a real-world foundation, rich semantic context, plan-aligned CoT, plus a modular structure for causal analysis. We select nuPlan as the foundation for DriveMind. As a large-scale planning benchmark released by Motional, nuPlan uniquely combines the high fidelity of real-world data with the rich semantics (e.g., traffic light status, lane topology) necessary for deep reasoning. We convert the raw log data from nuPlan into the final samples of the DriveMind dataset through an automated data generation pipeline, the core process of which is illustrated in Figure 1. Our data generation pipeline consists of three main stages, beginning with scene parsing and feature extraction, followed by GPT-4.1-based CoT generation, and concluding with human verification and final sample construction.

The first stage involves developing a parser to structure all key information from each nuPlan log. We extract ego-vehicle priors such as current and historical states, as well as the global navigation goal. For visual information, all camera images are first rescaled and then stitched into a three-row image grid corresponding to the front, side, and rear view groups to provide a spatially coherent input. We also encode lane information by identifying the ego-lane and its neighbors and determining their direction relative to the vehicle's heading, thereby enhancing the VLM's understanding of the road topology and its ability to identify drivable areas. Additionally, we process information on traffic signals by extracting the current state of relevant lights and the position of their corresponding stop lines. Finally, we extract features for dynamic objects within a 20-meter radius. We define four categories of valid targets: vehicles, pedestrians, bicycles, and other obstacles such as traffic cones or stone pillars. These targets are localized using 3D annotations from nuPlan ground truth, and we then employ GPT-4.1 to identify visual characteristics (e.g., 'a white SUV,' 'a pedestrian in a red jacket') of each object from their cropped images. This process enhances the agent's perception by creating an explicit link between each visual object and its corresponding textual description.

In the second stage, we use GPT-4.1 to generate high-quality CoT. Recognizing that current LLMs face challenges in precise 3D spatial reasoning from images alone Gao et al. (2025), we ground the reasoning process by providing GPT-4.1 with structured scene ground truth. We design a comprehensive multimodal prompt that includes the preprocessed visual input, the structured scene ground truth from the first stage, and the expert's future planning trajectory. GPT-4.1 is tasked with explaining the causal logic behind the expert trajectory. The resulting CoT is organized into a structured text with three parts, which are scene ground truth, a causal analysis, and a macro decision.

Finally, we perform human verification and construct the final samples. To ensure the quality of the automatically generated CoT, a random 10% (∼5K) of the samples are manually reviewed by human experts. This review process confirms a high degree of logical correctness in the generated reasoning. After that, all information is consolidated into a structured training sample. The model is provided with two inputs, the visual inputs containing the preprocessed images and the textual priors containing ego state, history, and navigation information. The model is trained to produce two target labels, the 'ground truth cot' representing the complete, human-verified CoT, and the 'ground truth planning', which is the expert driving trajectory. The 'ground truth planning' in the DriveMind dataset is derived entirely from the nuPlan human-expert driving logs, and GPT-4.1 is employed solely to generate the CoT part. The details of our reasoning generation process and an example including the complete input and target labels for a training sample are provided in Appendix A.2.

Our training set is constructed from 61 driving scenarios sourced from the nuPlan training split. To mitigate the inherent long-tail distribution of this real-world data, we employ square-root weighted stratified sampling. For a target dataset size of $M = 50,000$ across $K = 61$ scenarios, the number of samples $n_i$ drawn from the $i$-th scenario (which originally contains $N_i$ samples) is calculated as:

$$n_i = M \cdot \frac{\sqrt{N_i}}{\sum_{j=1}^{K} \sqrt{N_j}}, \tag{1}$$

By this sampling, our DriveMind dataset maintains a high degree of complexity with 33.9% hard scenarios (e.g., turnings and dense pedestrian interactions), 39.2% general scenarios (e.g., slow down at lights), while simple scenarios (e.g., stationary) comprise only 26.9%. This ensures that the model is trained primarily on scenarios requiring active reasoning rather than simple path following or being stationary. The detailed distribution of all 61 scenario types is provided in Appendix A.3

Table 1 provides a detailed comparison between DriveMind and other datasets. Besides the dense semantic context derived from the real-world nuPlan logs and the high-quality ground truth based CoT, the critical feature of our dataset is its modular information structure, which is designed specifically for causal analysis. By explicitly separating different priors within the coherent textual instructions,

Table 1: Comparison of datasets in terms of real-world logs, semantic richness, expert CoT availability, modularity, and scenario coverage.

| Dataset | Real-World Logs | Semantic Richness | Expert CoT | Modular Structure | Scenario Coverage |
|---|---|---|---|---|---|
| DriveLM | Mixed | Moderate (nuScenes), High (CARLA) | Yes | No | — |
| DriveVLM | Yes | Moderate (SUP-AD) | Yes | No | 40 |
| DriveCoT | No | High (CARLA) | Yes | No | 5 |
| OmniDrive | Yes | Moderate (nuScenes) | No | No | — |
| **DriveMind (Ours)** | **Yes** | **High (nuPlan)** | **Yes** | **Yes** | **61** |

Table 2: Overview of ablation settings for tuning analysis. By systematically removing priors, we conducted several ablation studies to examine the impact of CoT and driving priors on VLM-based driving agents. Scalability is further validated on Llava. GRPO-based experiments confirm that the reasoning-planning disconnect effect is intrinsic rather than a superficial artifact.

| Agent | Base Model | Tuning | Vision | CoT | Priors |
|---|---|---|---|---|---|
| *Supervised Fine-Tuning (SFT) Experiments* | | | | | |
| Base | Qwen2.5-vl-7b | — | ✓ | — | — |
| CoT | Qwen2.5-vl-7b | SFT | ✓ | ✓ | All |
| Plan | Qwen2.5-vl-7b | SFT | ✓ | ✗ | All |
| Plan_NoV | Qwen2.5-vl-7b | SFT | ✗ | ✗ | All |
| CoT_NoHis | Qwen2.5-vl-7b | SFT | ✓ | ✓ | w/o History |
| CoT_NoHis_Ego | Qwen2.5-vl-7b | SFT | ✓ | ✓ | w/o His, Ego |
| CoT_NoPri | Qwen2.5-vl-7b | SFT | ✓ | ✓ | None |
| CoT_L | llava1.6-mistral-7b | SFT | ✓ | ✓ | All |
| Plan_L_NoV | llava1.6-mistral–7b | SFT | ✗ | ✗ | All |
| Omnidrive | Omni | SFT | ✓ | Decomposed | All |
| Omnidive_NoPri | Omni | SFT | ✓ | Decomposed | None |
| *GRPO Policy Alignment Experiments* | | | | | |
| CoT_grpo | Qwen2.5-vl-7b | SFT+GRPO | ✓ | ✓ | All |
| Base_grpo | Qwen2.5-vl-7b | GRPO | ✓ | Self-learned | All |
| Base_grpo_NoPri | Qwen2.5-vl-7b | GRPO | ✓ | Self-learned | None |

researchers can perform precise information ablation. This allows for a reliable determination of whether an agent's planning stems from its autonomous visual perception and high-level reasoning, or from a reliance on "shortcut learning" from the provided textual priors.

### 3.2 TUNING VLMS FOR CAUSAL ANALYSIS

Since general-purpose VLMs are not pre-trained for driving planning, a crucial first step is to fine-tune them into proficient driving agents. Our primary objective, however, is to use the fine-tuning process itself as our object of study. We investigate how current training paradigms shape the causal structure of an agent's learned behaviors, for testing our Reasoning-Planning Decoupling Hypothesis. To this end, we conduct controlled experiments on a primary representative VLM, Qwen2.5-vl Bai et al. (2025), and verify the universality of our findings on a second architecture, Llava-1.6 Liu et al. (2023a). Crucially, our setting is made comprehensive by also including Omnidrive Wang et al. (2025). By analyzing Omnidrive, which employs counterfactual reasoning in an attempt to mitigate prior dependency, we can test our hypothesis against existing state-of-the-art solutions. This three-pronged approach, which covers in-depth analysis, architectural generalization, and state-of-the-art comparison, provides a robust and sufficient foundation for our claims.

We first employ SFT to create a set of agents under various information ablation conditions. At this stage, we utilized the DriveMind dataset and its variants shown in Table 2, where key input modalities, such as visual information, textual priors, or CoT are systematically removed. We fine-tune an agent with LoRA Hu et al. (2022) using each ablation dataset $D = \{((I_i, T_i), \mathcal{Y}_i)\}_{i=1}^{M}$, where $M$ is the total number of samples in the set. For each sample $i$, $I_i$ represents the visual input, $T_i$ is the textual prompt, and $\mathcal{Y}_i = (y_{i,1}, y_{i,2}, \ldots, y_{i,L})$ is the corresponding target output sequence of length $L$. With the pre-trained VLM parameters frozen as $\theta$ and the trainable matrices as $\phi$, we perform SFT. The cross-entropy loss is used to compare two probability distributions for output each token. The first is the model's predicted distribution ($\mathbf{q}$), which is the softmax output of the VLM. This is compared against the ground-truth distribution ($\mathbf{p}$), which is a one-hot vector where the probability is 1 for the correct token $y_{i,j}$ and 0 for all other tokens. The objective is to minimize the divergence between these two distributions, formulated as:

$$\mathcal{L}_{\text{SFT}}(\phi) = -\frac{1}{M} \sum_{i=1}^{M} \sum_{j=1}^{L} \sum_{v=1}^{V} \mathbf{p}_{i,j}(v) \log \mathbf{q}_{i,j}(v|I_i, T_i, y_{i,<j}; \theta, \phi). \quad (2)$$

To test the robustness of this disconnect finding, we then incorporate an RL-based policy alignment stage using GRPO, which explicitly enhances an agent's reasoning capabilities by rewarding the CoT process. We further sample $1,000$ new VQA samples from nuPlan, maintaining the same scenario distribution as DriveMind, and then do GRPO on both the SFT-trained model and the base model without SFT, with the latter serving as a baseline. Let $D_{grpo}$ denote the dataset and $\pi_\theta$ the policy, GRPO aims to maximize the following objective DeepSeek-AI et al. (2025):

$$\mathcal{J}_{\text{GRPO}}(\theta) = \mathbb{E}_{(I_j, T_j) \sim D_{grpo}, \{o_i\}_{i=1}^G \sim \pi_{\theta_{\text{old}}}} \left[ \tfrac{1}{G} \sum_{i=1}^G \mathcal{A}_i \, \tilde{w}_i - \beta \, \mathbb{D}_{\text{KL}}(\pi_\theta \| \pi_{\text{ref}}) \right]. \tag{3}$$

This objective means that for each pair of inputs, $G$ outputs are sampled from an older policy $\pi_{\theta_{\text{old}}}$, then each output is evaluated its normalized advantage $\mathcal{A}_i$ that measures its quality relative to the group's average. The $\tilde{w}_i$ is a weight to stabilizes updates by preventing overly large policy ratios and $\mathbb{D}_{\text{KL}}$ is a KL-divergence penalty to regularize the current policy $\pi_\theta$ to stay close to the base policy $\pi_{\text{ref}}$ with the tunable hyperparameter $\beta$. The model is encouraged to improve planning performance via enhanced reasoning by three rewards in $\mathcal{A}$: a location reward, a velocity reward, and a format reward. The first two rewards measure planning accuracy while the format reward incentivizes reasoning ability. For detailed explanations of GRPO and our reward functions, please refer to Appendix A.4. This stage serves as a critical test to demonstrate that the observed disconnect is not an artifact of the SFT process, but a persistent characteristic that remains even after a direct intervention designed to strengthen the reasoning-planning link.

### 3.3 Causal Diagnosing with Sequence-level Attention Analysis

To investigate our hypothesis and gain insight into the VLM's internal mechanisms, we employ an interpretability framework called Sequence-level Attention Analysis. The goal of it is to analyze the patterns of macro information flow between the reasoning and planning in the agent's output. The intuition is to follow the model's "gaze" as it generates its whole plan. When the model outputs trajectory tokens, we measure how much attention it pays to the preceding CoT tokens versus other contextual information (e.g., ego-state, history). A high degree of attention on the reasoning sequence would indicate a strong correlational link, suggesting the plan is at least conditioned on the reasoning. Conversely, low attention would provide evidence in line with our decoupling hypothesis.

We formalize this by analyzing the attention weights of the VLM, which has $L$ transformer layers and $H$ attention heads per layer. Let $a_{j \to i}^{(l,h)}$ denote the raw attention weight that the token at position $i$ (the query) assigns to the token at position $j$ (the key) in layer $l$ and head $h$. Our analysis then proceeds in two steps. First, we calculate a sequence-aggregated attention score, $\bar{a}_{j \to S_t}^{(l)}$. This score represents the total attention that a target sequence $S_t$ (e.g., the planning) directs to a single preceding token $j$, averaged over all $H$ attention heads in a given layer $l$:

$$\bar{a}_{j \to S_t}^{(l)} = \tfrac{1}{H} \sum_{i \in S_t} \sum_{h=1}^H a_{j \to i}^{(l,h)}. \tag{4}$$

Second, using this aggregated score, we compute our primary metric: the proportional attention score, $G_{S_M \to S_t}^{(l)}$. This value calculates the proportion of the target sequence's ($S_t$) total attention that is allocated to a specific source sequence $S_M$ (e.g., the CoT reasoning). Given a set of $M_{total}$ preceding source sequences $\{S_1, S_2, \ldots, S_{M_{total}}\}$, this attention proportion is given by:

$$G_{S_M \to S_t}^{(l)} = \left( \sum_{j \in S_M} \bar{a}_{j \to S_t}^{(l)} \right) / \left( \sum_{m=1}^{M_{total}} \sum_{j \in S_m} \bar{a}_{j \to S_t}^{(l)} \right). \tag{5}$$

This metric, which sums to 1 across all source sequences, provides a direct and quantitative measure of information dependency. By comparing the attention proportion directed towards the reasoning sequence versus other contextual sequences, we can rigorously verify whether an agent's planning is genuinely grounded in its own reasoning.

### 3.4 Causal Probe for VLM Driving Agents

Standard performance metrics (e.g., planning scores) are insufficient to distinguish between planning that originates from genuine reasoning and that relies on shortcut learning. To efficiently and deeply probe an agent's decision-making logic and diagnose its degree of reliance on shortcuts, we introduce a novel, training-free causal probe. The core principle of this probe is rooted in the concept of robustness: a planning decision genuinely driven by high-level reasoning from the visual

scene and CoT should be largely invariant to minor, semantically plausible perturbations in its textual priors. Conversely, an excessive reaction to such minor changes indicates a brittle reliance on those priors. Our framework introduces two distinct perturbation methods to test this principle.

The first method is a quantitative test we term *lateral offset perturbation*. Its underlying hypothesis is that a robust agent, truly reasoning based on the visual scene and CoT, should be able to ignore or correct for a small offset in its ego velocity because its visual input clearly reflects the vehicle's true position and heading in the world. In contrast, an agent merely pattern-matching the textual priors would be misguided. In this probe, we apply a small lateral offset, $\delta$, to the agent's ego-velocity. We then quantify the impact by measuring the Mean Final Deviation (MFD) (m) of the trajectory endpoint and the Relative Intervention Degree (RID). The MFD measures the absolute deviation:

$$\text{MFD} = \frac{1}{n} \sum_{i=1}^{n} |\text{Original\_Final\_xy}_i - \text{Perturbed\_Final\_xy}_i|, \tag{6}$$

where $n$ is the number of tested samples. RID is defined as the ratio of the difference between a sample's output before and after perturbation to the theoretical error that would occur if the model perfectly followed the perturbation (i.e., $\delta \cdot v_i \cdot T$):

$$\text{RID} = \frac{1}{n} \sum_{i=1}^{n} \frac{|\text{Original\_Final\_xy}_i - \text{Perturbed\_Final\_xy}_i|}{\delta \cdot v_i \cdot T}, \tag{7}$$

where $v_i$ is the ego-velocity of the $i$-th sample and $T$ is the prediction horizon. This normalized comparison enables RID to quantitatively measure the extent to which a model relies on textual priors: RID=1 indicates complete reliance on the error in textual priors, RID>1 indicates amplification of the error, and RID<1 indicates the ability to correct the error through visual reasoning. This provides a scale-independent, robust metric that allows for cross-scenario comparisons.

The second method, *lateral direction inversion*, is a qualitative probe. In this probe, we keep the ego-vehicle's current state (ego-state) unchanged but invert the lateral component of its historical trajectory (e.g., mirroring a history of "merging to the current position from the left" to one of "merging from the right"). The core hypothesis is that an agent reliant on shortcuts might misinterpret this inverted history and have its current and future decisions unduly influenced. A positive diagnosis for shortcut learning occurs when this inversion causes a directional change in the plan (e.g., planning a right turn based on the inverted history), while the generated CoT (which is primarily conditioned on the unchanged visual scene) still presents arguments for a left turn. This exposes a stark contradiction between the agent's stated reasoning and its actual planning logic.

By combining this quantitative measurement and qualitative diagnosis, we provide a systematic, training-free method to identify the presence and severity of shortcut learning in driving agents.

## 4 EXPERIMENT RESULTS AND ANALYSIS

### 4.1 EXPERIMENT SETUP

**Implementation Details:** Our experiments are conducted on two representative VLMs, Qwen2.5-vl and Llava-1.6, and include a reimplementation of the state-of-the-art Omnidrive method. We fine-tune the agents listed in Table 2, with the specific training parameters detailed in Appendix A.6.

**Evaluation Protocol:** For evaluation, we use the official nuPlan Challenge test set. To ensure comprehensive coverage, we randomly sample 200 scenarios for each of the 14 distinct types defined in the benchmark. We conduct both open-loop and closed-loop (non-reactive) evaluations for every agent to holistically assess the performance. More Details are provided in Appendix A.7.

### 4.2 TUNING ABLATION ANALYSIS RESULTS

Our primary finding, detailed in Table 3, is the VLM agent's profound reliance on textual priors over CoT reasoning for planning. The performance of fully-equipped agents that generate explicit reasoning, including both standard *CoT* and *Omnidrive*, is nearly indistinguishable from the *Plan* agent, which produces no reasoning output. This comparison strongly indicates that the articulated reasoning process offers no discernible benefit to the final planning outcome. The most definitive evidence for disconnect is a paradoxical result: the *Plan_NoV* agent, effectively 'driving blind' without any visual input, performs on par with the *CoT* agent. This is a powerful demonstration of shortcut learning. Since the priors contain no information about environmental conditions, it reveals the

Table 3: Main results of our ablation studies on the nuPlan test set, with all metrics reported at 1s, 2s, and 3s horizons. Our central finding is that the *Plan_NoV* agent performs nearly identically to the fully-equipped agents, demonstrating a profound reliance on textual priors. Performance collapses when all priors are removed (e.g., *CoT_NoPri*), confirming this dependency.

| Driving Agent | Open-Loop Score ↑ | | | Closed-Loop Score ↑ | | | Avg. ADE / FDE (m) ↓ | | | Collision Ratio ↓ | | |
|---|---|---|---|---|---|---|---|---|---|---|---|---|
| | 1s | 2s | 3s | 1s | 2s | 3s | 1s | 2s | 3s | 1s | 2s | 3s |
| Base | 36.52 | 33.62 | 31.50 | 59.80 | 42.48 | 36.99 | 2.67/4.45 | 4.80/8.81 | 7.02/13.75 | 7.84% | 9.16% | 9.87% |
| CoT | 98.91 | 96.67 | 92.56 | 97.46 | 96.42 | 92.38 | 0.03/0.08 | 0.14/0.39 | 0.33/1.02 | 0.00% | 0.29% | 2.15% |
| Omnidrive | 98.92 | 96.85 | 92.96 | 97.49 | 96.35 | 92.47 | 0.03/0.08 | 0.13/0.38 | 0.31/0.99 | 0.00% | 0.31% | 2.03% |
| Plan | 98.89 | 96.64 | 92.58 | 97.38 | 96.49 | 92.60 | 0.04/0.09 | 0.14/0.41 | 0.33/1.02 | 0.00% | 0.36% | 2.55% |
| Plan_NoV | 98.95 | 96.84 | 92.92 | 97.46 | 96.37 | 92.54 | 0.03/0.08 | 0.13/0.38 | 0.32/0.98 | 0.00% | 0.36% | 2.45% |
| CoT_NoHis | 97.44 | 89.30 | 82.82 | 96.71 | 94.17 | 87.93 | 0.11/0.27 | 0.38/0.98 | 0.78/2.08 | 0.12% | 1.62% | 5.09% |
| CoT_His_Ego | 74.51 | 61.65 | 57.70 | 90.39 | 87.23 | 81.32 | 0.69/1.25 | 1.35/2.60 | 2.04/4.05 | 0.24% | 1.80% | 5.27% |
| CoT_NoPri | 72.40 | 58.58 | 54.56 | 89.82 | 85.87 | 79.19 | 0.73/1.34 | 1.47/2.88 | 2.26/4.60 | 0.23% | 2.15% | 6.58% |
| Omnidrive_NoPri | 75.91 | 61.34 | 57.32 | 89.99 | 86.52 | 79.69 | 0.64/1.17 | 1.29/2.53 | 2.00/4.14 | 0.31% | 2.27% | 6.54% |
| CoT_L | 98.81 | 96.49 | 92.22 | 97.45 | 95.82 | 91.32 | 0.03/0.08 | 0.14/0.40 | 0.34/1.03 | 0.00% | 0.54% | 3.05% |
| Plan_L_NoV | 98.91 | 96.90 | 93.03 | 97.54 | 96.53 | 92.87 | 0.03/0.08 | 0.13/0.36 | 0.31/0.95 | 0.00% | 0.36% | 2.09% |
| Base_grpo | 96.38 | 85.31 | 77.84 | 96.74 | 92.52 | 84.51 | 0.15/0.35 | 0.47/1.20 | 0.95/2.50 | 0.12% | 1.74% | 6.22% |
| Base_grpo_NoP | 39.93 | 29.70 | 27.84 | 73.88 | 61.75 | 55.44 | 1.80/3.22 | 3.55/6.88 | 5.39/10.68 | 10.59% | 15.68% | 20.41% |
| CoT_grpo | 98.76 | 96.35 | 92.25 | 97.36 | 96.26 | 91.74 | 0.05/0.10 | 0.15/0.44 | 0.36/1.08 | 0.00% | 0.48% | 2.45% |

Table 4: Attention distributions across preceding sequences during target sequence generation.

| Attention Target | Scenario Reasoning Task | | | Planning Task | | |
|---|---|---|---|---|---|---|
| | Shallow Layer | Middle Layer | Final Layer | Shallow Layer | Middle Layer | Final Layer |
| Image Tokens (%) | **2.91** | **5.55** | **11.52** | 2.47 | 2.51 | 1.82 |
| Textual Priors (%) | 3.74 | 5.34 | 10.36 | **11.52** | **17.93** | **26.97** |
| Generated Reasoning (%) | — | — | — | 17.11 | 21.12 | 19.37 |
| All Textual Tokens (%) | 97.09 | 94.45 | 88.48 | 97.53 | 97.49 | 98.18 |

model has learned to even bypass scene understanding by merely extrapolating from its current state and history. Conversely, the removal of textual priors results in a catastrophic performance collapse, as evidenced by the sharp decline across all metrics for *CoT_NoPri* and *Omnidrive_NoPri*. The consistency of these findings across diverse models, including Qwen-VL, Llava-1.6, and Omnidrive, provides robust evidence for our Reasoning-Planning Decoupling Hypothesis.

To further subject our hypothesis to a rigorous stress test, we employ GRPO, a powerful policy alignment tool. In theory, by rewarding high-quality planning outcomes (defining expert trajectories as the positive preference), GRPO has the potential to optimize the entire upstream cognitive chain, making it the most promising method for forging a causal link between reasoning and planning. However, our experiment results directly contradict this optimistic outlook. While the *Base_grpo* agent shows superficial improvements over the *Base* counterpart, this proves to be a "false prosperity" built on shortcuts. Once textual priors are removed, the *Base_grpo_NoPri* agent's performance collapses catastrophically, falling well below even the original *Base* model. This decisively shows that this powerful alignment tool does not guide the model to learn deeper visual reasoning; instead, it rewards and reinforces its reliance on textual shortcuts, exacerbating the overfitting problem. Furthermore, the *CoT_grpo* agent's planning capability is even weaker than that of *CoT*, suggesting that forcibly aligning a disconnected reasoning process with planning can be counterproductive. Ultimately, this stress test provides our most compelling support for the conclusion that the reasoning-planning disconnect is a deeply ingrained characteristic of the current VLM fine-tuning paradigm. Even advanced policy optimization designed to solve alignment issues fails to rectify this disconnect and may even intensify the model's dependence on shortcuts.

### 4.3 SEQUENCE-LEVEL ATTENTION ANALYSIS RESULTS

To investigate the agent's decision-making process, we analyze the attention flow of the *CoT_grpo* agent. We partition the model's output into two key sequences: the *Reasoning* (CoT) sequence and the *Planning* (trajectory) sequence. We then measure the proportion of attention each sequence directs towards different preceding information sources (Images, Textual Priors, etc.) at shallow, middle, and final layers of the model, allowing us to observe how information dependencies evolve.

Table 5: Impact of different perturbation intensities ($\delta$) on *CoT_grpo*. The larger MFD and RID demonstrate its strong reliance on text priors.

|              | 0.02 | 0.06 | 0.08 | 0.10 | 0.30  |
|--------------|------|------|------|------|-------|
| **MFD** ↓   | 0.55 | 2.17 | 3.75 | 5.89 | 15.46 |
| **RID** ↓   | 1.00 | 1.47 | 2.04 | 2.17 | 2.04  |

As shown in Table 4, when the agent generates the *Reasoning* sequence, its attention to the Image Tokens progressively increases, from $2.91\%$ in the shallow layer to over $11\%$ in the final layer. This indicates that the reasoning process is continuously and increasingly grounded in the visual input. While attention to Textual Priors also grows, it remains secondary to the visual information in deeper layers. This behavior represents a healthy, logical process where the agent builds its reasoning primarily upon what it sees, while integrating priors as necessary context.

In contrast, the attention shifts dramatically during the *Planning* phase. Attention to the model's generated Reasoning first increases and then fades, while attention to the Textual Priors skyrockets from $11.52\%$ to nearly $27\%$ in the final layer. This shows the model discovering the decisive role of priors and becoming highly dependent on them. Simultaneously, attention to the Image Tokens becomes negligible, dropping below $2\%$, which indicates that the model barely requires visual information for planning. This finding, combined with the high performance of the *Plan_NoV* agent in Table 3, leads to an unambiguous conclusion: the VLM driving agent relies almost exclusively on textual priors for planning, rendering the reasoning and planning processes almost entirely disconnected.

A consistent observation across both tasks is the VLM's overwhelming preference for attending to textual tokens over image tokens. This highlights an inherent modality bias in current VLMs. This insight suggests that a potential path to mitigating shortcut learning in VLM driving agents is to enhance their visual understanding capabilities, forcing them to rely more on the visual modality.

### 4.4 CAUSAL PROBE DETECTION RESULTS

To evaluate the diagnostic efficacy of our causal probe, we select two key agents for this analysis: the strongest reasoning agent, *CoT_grpo*, and a state-of-the-art baseline known to include measures against prior dependency, *Omnidrive*. First, we apply the *lateral offset perturbation*. In this test, we add a small lateral velocity, $\delta$, perpendicular to the vehicle's heading, scaled to the ego-vehicle's current speed $v_{\text{ego}}$ (i.e., $\delta = 0.1 \cdot v_{\text{ego}}$). This creates a small and semantically plausible perturbation. As shown in Figure 2, both agents exhibit extreme sensitivity to this dynamic perturbation.

For more precise quantitative analysis and to test the model's sensitivity under different perturbation levels, we test on the *CoT_grpo* utilizing a test set of 300 randomly sampled scenarios with varied the perturbation magnitude $\delta$ from 0.02 to 0.3, and we report the MFD and RID in Table 5. From the results, the model exhibits extreme instability even at low perturbation levels. As $\delta$ increases from 0.02 to 0.06, the MFD jumps significantly to 2.17 m. This confirms that the agent is captured by the textual prior almost immediately, lacking the robustness to correct for even subtle semantic conflicts. As $\delta$ increases further the MFD continues to grow monotonically, reaching 15.46m at $\delta = 0.3$. The RID initially peaks at 2.17 for $\delta = 0.1$, then plateaus or slightly decreases to 2.04 for $\delta = 0.3$. Actually, this plateau does not indicate robustness as the RID is still larger than 1; rather, it reflects saturation of "wrong outputs" as the agent predicts extreme trajectories beyond its learned manifold, lagging behind increasing perturbations. The disproportionate reactions of the driving agents under small lateral offset perturbation from both the qualitative and quantitative results strongly indicate the agents' heavy reliance on the ego states in text priors for shortcut learning.

Next, we apply the *lateral direction inversion* probe. After inverting the lateral component of the historical trajectory, we observed that both *CoT_grpo* and *Omnidrive* consistently reversed their planned maneuver direction. Most critically, a manual inspection of the *CoT_grpo* agent's outputs after this inversion revealed a direct contradiction in 100% of the tested samples. In these cases, the agent would generate a correct CoT based on the visual scene (e.g., reasoning for a left turn) but then produce a plan that incorrectly followed the perturbed history (e.g., executing a right turn).

This finding provides definitive evidence for our Reasoning-Planning Decoupling Hypothesis. It not only demonstrates the deep-rooted nature of the problem but also validates the efficacy of our causal probes in uncovering these critical failures in reasoning. More examples are shown in Appendix A.9.

| Front Image | Model | No Perturbation | Lateral Offset Perturbation | Lateral Direction Inversion |
|---|---|---|---|---|
| | *CoT_grpo* | Reasoning: Given the presence of a left curve...**steer more sharply to the left**... ✔ Planning: (1.78, **0.07**),(9.07, **0.99**),(18.44, **3.62**)... ✔ | Reasoning: Given the presence of a left curve...likely **continue steering left**... ✔ Planning: (1.78, **6.43**),(8.84, **9.19**), (17.29, **13.91**)... ✘ | Reasoning: Given the presence of a left curve...**steer more sharply to the left**... ✔ Planning: (1.78, **-0.04**),(8.84, **-0.39**), (17.29, **-1.37**)... ✘ |
| | *Omnidrive* | Planning: (1.78, **0.06**),(9.03, **0.80**), (18.33, **3.54**)... ✔ | Planning: (1.78, **0.60**),(8.91, **3.98**), (17.42, **10.51**)... ✘ | Planning: (1.78, **-0.04**),(9.10, **-0.40**), (18.67, **-1.06**)... ✘ |
| **Plannings from *CoT_grpo*** | | **Planning Within Tolerance** | **Large Planning Deviation** | **Inversion of Lateral Direction** |

Figure 2: An illustration of our causal probe's diagnostic capabilities. The figure contrasts an agent's original plan with its divergent trajectories when subjected to two types of perturbations

## 4.5 EVALUATION ON LARGER MODELS

To assess whether our findings generalize to larger VLMs, we apply the same causal probes to the substantially larger GPT-4o in a zero-shot setting. Under perturbation, GPT-4o produced correct scenario reasoning (e.g., "The ego-vehicle is in a lane moving straight"), yet its predicted trajectory still adhered to the falsified textual prior, indicating that the reasoning–planning disconnect persists beyond the 7B model scale. Quantitatively, with a perturbation level of $\delta = 0.1$, GPT-4o yields an MFD of 2.55 m and an RID of 1.35. This zero-shot evaluation provides a necessary assessment of a closed-source model, demonstrating that the disconnect is fundamental to both open-source and proprietary large-scale VLM architectures.

## 5 FUTURE WORK

Our future work will focus on resolving the reasoning-planning disconnect. We plan to pursue two research directions aimed at developing a more robust and causally-faithful VLM driving agent.

**Mitigating Modality Bias via Contrastive Pre-Finetuning.** Our main results revealed a key finding: an agent with no visual input (*Plan_NoV*) can achieve planning scores nearly identical to a fully multimodal agent. This strongly suggests that the models have learned to almost entirely disregard the visual modality. Therefore, a critical future direction is to introduce a dedicated contrastive pre-finetuning stage designed to make the visual input indispensable. In this stage, we will train the VLM on a curated, driving-related dataset where the textual input is held constant while the visual input varies to produce a different correct output. This process will force the model to extract decisive information from the visual modality, rebalancing its modality dependence and enhancing the salience of the vision encoder before the downstream driving SFT/GRPO begins.

**Breaking Shortcuts via Contrastive Learning.** Our second direction is to enhance the agent's SFT/GRPO process via contrastive learning, leveraging negative examples to guide its policy. This involves augmenting the training data with verified conflict samples. For a target scene (e.g., Image_A), we pair it with a mismatched textual prior (e.g., Priors_B) from another scenario whose original ground-truth trajectory (Plan_B) has first been verified via the nuPlan metrics as an unsafe plan for the target scene. For these augmented samples, we will implement a repulsive loss function. This objective penalizes the policy if its predicted trajectory falls within a certain distance margin of the undesirable trajectory, Plan_B. The agent is forced to abandon the "prior equals plan" shortcut and instead learn to genuinely reason, aiming to connect the reasoning and planning.

Furthermore, these two directions are highly complementary. Mitigating modality bias aims to improve the information flowing into the reasoning module, while the contrastive learning strategy aims to enforce that the plan is correctly derived from it. Their combination promises a driving agent whose planning is grounded in its own reasoning.

## 6 CONCLUSION

In this paper, we have investigated the causal link between reasoning and planning in VLM driving agents. To enable this analysis, we have created the DriveMind dataset, a benchmark designed for causal analysis, and have developed a suite of diagnostic tools including controlled ablations and perturbation-based probes. Our experiments have provided strong, multi-faceted evidence supporting the Reasoning-Planning Decoupling Hypothesis. We have found that agents learn to shortcut, relying predominantly on textual priors for planning, while their generated CoT often serves as a plausible but non-causal byproduct. This work has revealed that the perceived interpretability of current agents can be misleading, as the reasoning are not causally linked to the final action, and has underscored the need for causally-aware training paradigms to build truly robust driving agents.

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

# A  APPENDIX

## A.1  THE USE OF LARGE LANGUAGE MODELS (LLMS)

In preparing this manuscript, we utilized a LLM (i.e., GPT-5) as a tool to enhance the clarity and precision of our writing. The core ideas, arguments, and scientific contributions were conceived and authored by the human writers. The LLM was employed to refine sentence structure, improve readability, and ensure grammatical correctness on initial human-written drafts. The final version of the manuscript reflects the authors' thorough review and final judgment on all text.

## A.2  EXAMPLE DATA OF DRIVEMIND DATASET AND GENERATION DETAILS

This section offers a detailed overview of the DriveMind dataset, covering both its structure and its generation process.

Figure 6 illustrates an example of data from the DriveMind dataset. Our input consists of concatenated surround views and textual inputs (including prior knowledge and instructions). The target output is structured into two parts. First, a plan-aligned CoT is presented within a `<think>` block. This CoT incorporates a comprehensive analysis of the scene, including key static elements (e.g., lanes, traffic lights, road signs), environmental conditions (e.g., weather), and critical dynamic road users (e.g., nearby vehicles, pedestrians, and bicycles), concluding with a macro-level driving decision. Second, following this reasoning, the final trajectory plan is provided within an `<answer>` block. All the reasoning and planning are causally linked through the macro driving decision and the explicit causal logic-enhancing sentence. We highlight in green the information discussed in the main text that may contribute to shortcut learning: ego state, historical data, and navigation information. Our ablation experiments in the main text confirm that VLM driving agents' planning indeed originates from shortcut learning of textual priors, with reasoning emerging as a logical byproduct.

To generate the CoT data, we first use the info parser to extract and make the structured driving info from the nuPlan ground truth log. Then, we use the prompts illustrated in Figure 5 to instruct the GPT-4.1 to generate scenario-aware analysis of the target part, such as weather and detected vehicles. We input the images and corresponding ground truth from the structured info for each part to address the visual-spatial perception limitations inherent in GPT-4.1, and let it generate analyses for each part in sequence. When generating the macro driving decision, all preceding environmental ground truth perceptions, analyses, and the future trajectory are incorporated as inputs, with the given future trajectory ensuring the accuracy of the decision. Notably, to ensure the causality of the reasoning towards driving, we imposed the following constraints on GPT-4.1 when letting it give the macro driving decision: 'While the future trajectory is known, your tone should remain predictive: extrapolating logically from the present and can not report the known future data in your driving decision. Use inferential language, but do NOT use non-predictive language like: according to the future. Your decision must be grounded in present state logic and reflect the causal relationships between the current environment and the driving decision. Remember to remain predictive when extrapolating.' These operations ensure both the accuracy and the causality of our macro driving decision. After all the reasoning generated, we combine them with the ground truth information, plus a causal logic-enhancing sentence, to form the CoT data. Finally, a portion of the data will undergo rigorous manual verification to ensure content and logical accuracy.

## A.3  DETAILS OF THE SCENARIO DISTRIBUTION OF DRIVEMIND DATASET

We report the exact number of samples associated with each scenario category in the DriveMind dataset, organized by difficulty level:

- Hard: High-dynamic scenarios involving complex interactions (e.g., turning, lane changing).

- General: Standard driving scenarios governed by clear traffic rules (e.g., following lane with lead, stopping at lights).

- Simple: Low-dynamic or static scenarios (e.g., stationary in traffic, low magnitude speed).

The corresponding proportions of scenarios across these difficulty levels are also provided. Our square-root–weighted stratified sampling strategy (see Eq. (1)) yields a well-balanced distribution across difficulty levels, resulting in 33.9% Hard, 39.2% General, and 26.9% Simple scenarios. Detailed statistics are presented in Table 6.

Table 6: Distribution of scenario types and difficulty-level proportions in the DriveMind.

| Difficulty Category | Scenario Name | Sample Count | Total/Proportion |
|---|---|---|---|
| Hard | changing_lane_to_left | 56 | 16935/33.9% |
| | changing_lane_to_right | 51 | |
| | changing_lane_with_trail | 24 | |
| | crossed_by_vehicle | 41 | |
| | high_lateral_acceleration | 648 | |
| | starting_protected_cross_turn | 291 | |
| | starting_protected_noncross_turn | 434 | |
| | high_magnitude_jerk | 33 | |
| | near_barrier_on_driveable | 274 | |
| | near_construction_zone_sign | 1033 | |
| | near_long_vehicle | 1739 | |
| | near_multiple_pedestrians | 395 | |
| | near_pedestrian_at_pickup_dropoff | 1474 | |
| | near_pedestrian_on_crosswalk | 2619 | |
| | near_pedestrian_on_crosswalk_with_ego | 267 | |
| | near_trafficcone_on_driveable | 1251 | |
| | starting_high_speed_turn | 206 | |
| | starting_unprotected_cross_turn | 337 | |
| | starting_unprotected_noncross_turn | 275 | |
| | traversing_narrow_lane | 26 | |
| | behind_long_vehicle | 711 | |
| | behind_pedestrian_on_driveable | 69 | |
| | behind_pedestrian_on_pickup_dropoff | 26 | |
| | behind_bike | 205 | |
| | near_multiple_vehicles | 1363 | |
| | traversing_pickup_dropoff | 3087 | |
| General | near_high_speed_vehicle | 1204 | 19626/39.2% |
| | waiting_for_pedestrian_to_cross | 301 | |
| | accelerating_at_crosswalk | 137 | |
| | accelerating_at_stop_sign_no_crosswalk | 84 | |
| | accelerating_at_stop_sign | 152 | |
| | accelerating_at_traffic_light | 100 | |
| | accelerating_at_traffic_light_with_lead | 115 | |
| | accelerating_at_traffic_light_without_lead | 357 | |
| | following_lane_with_lead | 214 | |
| | following_lane_with_slow_lead | 828 | |
| | on_all_way_stop_intersection | 918 | |
| | on_intersection | 1142 | |
| | on_stopline_crosswalk | 1014 | |
| | on_stopline_stop_sign | 788 | |
| | on_stopline_traffic_light | 1342 | |
| | on_traffic_light_intersection | 2029 | |
| | starting_low_speed_turn | 141 | |
| | starting_straight_stop_sign_intersection_traversal | 323 | |
| | starting_straight_traffic_light_intersection_traversal | 364 | |
| | stopping_at_crosswalk | 219 | |
| | stopping_at_stop_sign_no_crosswalk | 100 | |
| | stopping_at_stop_sign_without_lead | 166 | |
| | stopping_at_traffic_light_with_lead | 75 | |
| | stopping_at_traffic_light_without_lead | 122 | |
| | stopping_with_lead | 219 | |
| | traversing_crosswalk | 1665 | |
| | traversing_intersection | 1926 | |
| | traversing_traffic_light_intersection | 3581 | |
| Simple | stationary_at_crosswalk | 63 | 13440/26.9% |
| | stationary_at_traffic_light_with_lead | 1206 | |
| | stationary_at_traffic_light_without_lead | 1680 | |
| | low_magnitude_speed | 1788 | |
| | medium_magnitude_speed | 3547 | |
| | following_lane_without_lead | 1388 | |
| | high_magnitude_speed | 3768 | |

## A.4 GRPO FOR VLM DRIVING AGENTS

This section provides the detailed formulation of the GRPO implementation. We employ three rewards, a location reward, a velocity reward, and a format reward, for our GRPO training. The first two are calculated from the L2 error between each predicted point and its ground truth point, while the latter is consistent with that used in DeepSeek-AI et al. (2025). Note the $K = 3$ rewards as $r_1$, $r_2$, $r_3$ with their weights $\alpha_1$, $\alpha_2$, and $\alpha_3$, the normalized advantage $\mathcal{A}_i$ for an output in one group is calculated as:

$$\mathcal{A}_i = \frac{R_i - \text{mean}(R_1, \dots, R_G)}{\text{std}(R_1, \dots, R_G)}, \quad \text{where} \quad R_i = \sum_{k=1}^{K} \alpha_k \cdot r_k. \tag{8}$$

Besides, in equation 3, the $\tilde{w}_i$ is calculated as:

$$\tilde{w}_i = \min \left( \frac{\pi_\theta(o_i | (I_j, T_j))}{\pi_{\theta_{\text{old}}}(o_i | (I_j, T_j))}, \text{clip} \left( \frac{\pi_\theta(o_i | (I_j, T_j))}{\pi_{\theta_{\text{old}}}(o_i | (I_j, T_j))}, 1 - \epsilon, 1 + \epsilon \right) \right), \tag{9}$$

where the $\epsilon$ is a tunable hyperparameter. The KL-divergence penalty item $\mathbb{D}_{\text{KL}}$ is calculated as:

$$\mathbb{D}_{\text{KL}}(\pi_\theta || \pi_{\text{ref}}) = \frac{\pi_{\text{ref}}(o_i | (I_j, T_j))}{\pi_\theta(o_i | (I_j, T_j))} - \log \frac{\pi_{\text{ref}}(o_i | (I_j, T_j))}{\pi_\theta(o_i | (I_j, T_j))} - 1. \tag{10}$$

## A.5 TUNING ABLATION DETAILS

Our tuning ablation experiments are based on our proposed dataset, DriveMind, which contains samples with modular ground truth input and detailed plan-aligned CoT reasoning towards the planning task. We conduct extensive data-driven ablations on different models, verifying the disconnect between reasoning and planning and the shortcut learning leveraging text priors in VLM driving agents. The detailed supplementary information of our ablations is shown in Table 7.

## A.6 TRAINING DETAILS FOR AGENTS

In this work, we conduct standard SFT and GRPO training on our agents based on the LLM training and development framework, Scalable Lightweight Infrastructure for Fine-Tuning (SWIFT) Zhao et al. (2024). For the SFT, we employ LoRA with rank and alpha both set to 64. Training is conducted on the aligner, ViT, and LLM backbone, using learning rates of 1e-4, 1e-5, and 1e-5, respectively. The batch size is set to 8, with a warm-up ratio of 0.05. Cosine learning rate scheduling is applied to achieve smoother training. For the GRPO training, we employ the same settings, plus the temperature parameter for the policy model that generates samples is set to 0.9 to encourage a degree of exploration, the group size is set to 8, and the warm-up ratio is set to 0.01. For the three rewards, we assign weights of 0.45, 0.45, and 0.1 to the location, velocity, and format rewards for *CoT_grpo*, respectively, whereas for *Base_grpo*, the three rewards are equally weighted.

## A.7 NUPLAN CHALLENGE METRICS

This section provides detailed definitions for the nuPlan Challenge Motional (2023) evaluation metrics used in our experiments.

### A.7.1 OPEN-LOOP METRICS

In each scenario, the agent's planned trajectory is compared against the expert's ground-truth trajectory over 1s, 2s, and 3s horizons. The comparison is based on the following five core metrics, which are first computed at each sampled timestep and then averaged over all timesteps in the scenario.

**Average Displacement Error (ADE).** At each timestep, ADE is the mean of the pointwise L2 distances between the planned and expert trajectories over the selected future horizon.

**Final Displacement Error (FDE).** At each timestep, FDE is the L2 distance between the planned and expert trajectories at the end of the selected future horizon.

**Average Heading Error (AHE).** At each timestep, AHE is the mean of the absolute differences in heading angle between the planned and expert trajectories over the future horizon.

**Final Heading Error (FHE).** At each timestep, FHE is the absolute difference in heading angle at the end of the future horizon.

**Miss Rate.** At each timestep, a "miss" is declared if the maximum pointwise L2 distance between the planned and expert trajectories exceeds a threshold, which is defined as 2.0m for the 1s horizon,

Table 7: Details of our tuning ablation analysis settings.

| Agent | Ablation Details with DriveMind |
|---|---|
| Base | Base model without any finetuning. Serves as a control group. |
| CoT | Finetuned on the full DriveMind dataset with ∼50K CoT VQA samples. |
| Plan | The reasoning part is removed from each sample in DriveMind, retaining only the final planning results. |
| Plan_NoV | Same as 'Plan', but all visual inputs are also removed. |
| CoT_NoHis | Ego's history information is removed from the textual input of the samples in DriveMind. |
| CoT_NoHis_Ego | Both history and the ego state are removed from the textual input of the samples in DriveMind. |
| CoT_NoPri | All driving priors, history, ego state, and navigation, are removed from the textual input. |
| CoT_L | Scalability test: Replicates the 'CoT' setup on the LLaVA-1.6. |
| Plan_L_NoV | Scalability test: Replicates the 'Plan_NoV' setup on the LLaVA-1.6. |
| Omnidrive | CoT samples from the DriveMind are decomposed into sub-tasks(detection, planning, etc.) without plan-aligned reasoning. Use all the decomposed sub-task data to simultaneously train the Omnidrive agent. |
| Omnidrive_NoPri | Same as 'Omnidrive', with all the priors removed from text input for each sample. |
| CoT_grpo | The 'CoT' model is further tuned with GRPO using the $1,000$ grpo VQA samples that have the same input settings as DriveMind. |
| Base_grpo | The base model is directly tuned with GRPO using the $1,000$ grpo VQA samples. |
| Base_grpo_NoPri | Same as 'Base_grpo', but all priors in the samples are removed from the text input during GRPO tuning. |

3.2m for 2s, and 6.0m for 3s. The scenario's Miss Rate is the ratio of "missed" timesteps to the total number of timesteps.

**Open-Loop Score.** The final Open-Loop Score reported in the paper is a composite metric derived from the core metrics via a two-stage process. First, the scenario-level average of each of the five core metrics is compared against a predefined threshold to generate five binary scores ($S_{\text{metric}} \in \{0, 1\}$, where 1 represents a pass). A failure score of 0 is assigned if the scenario's Miss Rate is greater than 30%, if the average ADE or FDE is greater than 8.0 meters, or if the average AHE or FHE is greater than 0.8 radians. These binary scores are then aggregated into a final score ranging from 0 to 100. The $S_{\text{MissRate}}$ score acts as a gate; if it is 0, the final score for the scenario is 0. Otherwise, the final score is a weighted average of the other four binary scores, calculated as:

$$\text{Score} = \frac{2 \cdot S_{\text{ADE}} + 2 \cdot S_{\text{FDE}} + S_{\text{AHE}} + S_{\text{FHE}}}{6} \times S_{\text{MissRate}} \times 100 \tag{11}$$

### A.7.2 CLOSE-LOOP METRICS

The Closed-Loop Score evaluates the agent's ability to drive safely, adhere to traffic rules, and maintain comfort in an interactive simulation. The score is a composite metric calculated from eight core metrics using a hybrid gating and weighted-average formula. These metrics are divided into two main groups, Gating Metrics and Weighted Metrics.

The first group consists of three gating metrics, which represent critical safety and progress requirements. A failure in any of these results in an immediate scenario score of 0.

**No At-Fault Collisions (Collision Ratio).** This metric is a three-tiered score. It is assigned a value of 0 for an at-fault collision with a vehicle or Vulnerable Road User (VRU), or for multiple at-fault collisions with objects. It is 0.5 for a single at-fault collision with an object, and 1 otherwise. A collision is considered "at-fault" if it should have been preventable by the planner, such as colliding with a stopped object or a frontal collision.

**Drivable Area Compliance.** This is a binary score that becomes 0 if, at any frame, the maximum distance of any corner of the ego's bounding box from the nearest drivable area is more than 0.3m. This tolerance accounts for potential over-approximations of the ego bounding box.

**Making Progress.** This score is a binary value that is 0 if the agent's progress ratio along the expert route falls below the min progress threshold of 0.2.

The second group comprises five weighted metrics, which are scored between 0 and 1 and contribute to the final performance score via a weighted average.

Table 8: Attention distributions for the *Omnidrive* across preceding sequences during target sequence generation.

| Attention Target | Detection Task | | | Planning Task | | |
|---|---|---|---|---|---|---|
| | Shallow Layer | Middle Layer | Final Layer | Shallow Layer | Middle Layer | Final Layer |
| Image Tokens (%) | **3.46** | **10.02** | **15.92** | 1.96 | 3.22 | 4.05 |
| Textual Priors (%) | 6.76 | 5.93 | 6.89 | **10.52** | **9.93** | **23.97** |
| Textual Tokens (%) | 96.54 | 89.98 | 84.08 | 98.04 | 96.78 | 95.95 |

**Driving Direction Compliance.** This score is based on the distance traveled against traffic flow over a 1s time horizon; it is 1 if this distance is less than 2m, 0 if it is more than 6m, and 0.5 otherwise.

**Time to Collision (TTC).** This score is a binary value based on the minimum TTC, which is calculated by projecting bounding boxes forward up to a 3.0s horizon. The score is 0 if the minimum TTC falls below the least TTC safety threshold of 0.95s, and 1 otherwise.

**Speed Limit Compliance.** This metric evaluates the agent's adherence to posted speed limits from the map data, penalizing both the magnitude and duration of over-speeding. This score is a continuous value between 0 and 1, calculated as: $S_{\text{speed}} = \max\left(0, 1 - \frac{v_{\text{int}}}{v_{\text{thresh}} \cdot T_{\text{scenario}}}\right)$, where $v_{\text{int}}$ is the time-integral of the speed violation (i.e., the area under the over-speed vs. time graph), $T_{\text{scenario}}$ is the total duration of the scenario, and $v_{\text{thresh}}$ is the maximum acceptable over-speeding threshold, set to 2.23 m/s ($\approx$ 5 mph). A score of 1 indicates no violations, and the score trends towards 0 as the severity and duration of over-speeding increase.

**Ego Progress along Expert Route.** This metric measures how effectively the agent advances along the path taken by the expert driver. At each timestep, the agent's movement is projected onto this path to calculate its per-frame progress. The total progress for both the agent, $P_{\text{ego}}$, and the expert, $P_{\text{expert}}$, is then calculated by integrating these per-frame values over the entire scenario. This score is the ratio of these two values, capped between 0 and 1, and is calculated as: $S_{\text{progress}} = \min\left(1, \frac{\max(P_{\text{ego}}, \tau)}{\max(P_{\text{expert}}, \tau)}\right)$, where $\tau$ is a small threshold (0.1m) used to handle minor negative progress values that can arise from data noise and to prevent division by zero in scenarios with no movement. A score of 1 indicates the agent made at least as much progress as the expert.

**Comfort.** This metric quantifies the smoothness of the planned trajectory by ensuring key variables remain within bounds representative of a comfortable human driving experience. At each timestep of the scenario, a set of quantities are checked against empirically determined thresholds derived from an analysis of expert human driver trajectories. The scenario is deemed uncomfortable, and the score $S_{\text{comfort}}$ is set to 0, if *any* of the following conditions are violated at *any* time: longitudinal acceleration is outside the range of [-4.05, 2.40] m/s$^2$; absolute lateral acceleration exceeds 4.89 m/s$^2$; absolute yaw rate exceeds 0.95 rad/s; absolute yaw acceleration exceeds 1.93 rad/s$^2$; absolute longitudinal jerk exceeds 4.13 m/s$^3$; or the magnitude of the jerk vector exceeds 8.37 m/s$^3$. If none of these bounds are violated throughout the entire scenario, the score $S_{\text{comfort}}$ is 1.

**Close-Loop Score.** The final Closed-Loop Score is the product of a gating factor ($S_{\text{gate}}$) and a weighted performance score ($S_{\text{weighted}}$). First, the gating factor is the product of the three binary gating metrics, ensuring all critical safety rules are met:

$$S_{\text{gate}} = S_{\text{no\_collision}} \times S_{\text{drivable\_area}} \times S_{\text{making\_progress}} \tag{12}$$

Second, the weighted performance score is the weighted average of the five weighted metrics, with a total weight of 21:

$$S_{\text{weighted}} = \frac{5S_{\text{direction}} + 5S_{\text{TTC}} + 4S_{\text{speed}} + 5S_{\text{progress}} + 2S_{\text{comfort}}}{21} \tag{13}$$

The final Closed-Loop Score is then calculated as follows, where if $S_{\text{gate}}$ is 0, the total score is 0:

$$\text{Score} = S_{\text{gate}} \times S_{\text{weighted}} \times 100 \tag{14}$$

A.8 SEQUENCE-LEVEL SEGMENT ATTENTION ANALYSIS FOR OMNIDRIVE

We also conduct the sequence-level attention analysis on the *Omnidrive* model. We primarily select the object detection task (reflecting implicit reasoning processes) and the planning task from the

| Front Image | Model | No Perturbation | Lateral Offset Perturbation - Left | Lateral Offset Perturbation - Right |
|---|---|---|---|---|
| | CoT_grpo | Reasoning: maneuver involves progressing **forward with increasing velocity**... ✓ Planning: (1.45, **0.00**),(7.81, **-0.01**), (16.87,**-0.06**)... ✓ | Reasoning: **acceleration** and minor steering adjustments to **remain centered**... ✓ Planning: (1.45, **0.09**),(7.82, **0.94**), (16.80, **2.97**)... ✗ | Reasoning: continue **accelerating smoothly** through the intersection... ✓ Planning: (1.45, **--0.14**),(7.73, **-1.19**), (16.52, **-3.27**)... ✗ |
| | Omnidrive | Planning: (1.46, **0.00**),(7.81, **0.00**), (16.67, **0.00**)... ✓ | Planning: (1.45, **0.11**),(7.70, **0.83**), (16.21, **2.71**)... ✗ | Planning: (1.45, **-0.10**),(7.79, **-1.07**), (18.67, **-3.09**)... ✗ |
| Plannings from CoT_grpo | | **Planning Within Tolerance** | **Large Planning Deviation** | **Large Planning Deviation** |

Figure 3: An illustration of our causal probe's diagnostic capabilities in a straight driving scenario. The figure contrasts an agent's original plan with its divergent trajectories when subjected to lateral offset perturbations to the left and right. The lateral trajectories of both models' output undergo a significant change in response to the perturbations.

| Front Image | Model | No Perturbation | Lateral Offset Perturbation - Left | Lateral Offset Perturbation - Right |
|---|---|---|---|---|
| | CoT_grpo | Reasoning: red traffic light...multiple pedestrians actively...**remain stopped**... ✓ Planning: (0.00, **0.00**),(0.00, **0.00**), (0.00, **0.00**)... ✓ | Reasoning: red traffic light, a stationary lead vehicle...**remain stopped**... ✓ Planning: (0.00, **0.15**),(0.00, **1.49**), (0.00, **3.15**)... ✗ | Reasoning: red traffic light...pedestrians actively crossing...**remain stopped**... ✓ Planning: (0.00, **-0.12**),(0.00, **-1.39**), (0.00, **-2.97**)... ✗ |
| | Omnidrive | Planning: (0.00, **0.00**),(0.00, **0.00**), (0.00, **0.00**)... ✓ | Planning: (0.00, **0.19**),(0.00, **1.52**), (0.00, **3.17**)... ✗ | Planning: (0.00, **-0.09**),(0.00, **-1.24**), (0.00, **-2.86**)... ✗ |
| Plannings from CoT_grpo | | **Planning Within Tolerance** | **Large Planning Deviation** | **Large Planning Deviation** |

Figure 4: An illustration of our causal probe's diagnostic capabilities in a stopping scenario. The figure contrasts an agent's original plan with its divergent trajectories when subjected to lateral offset perturbations to the left and right. Models output distinct lateral 'translation' trajectories after perturbation, fully demonstrating the disconnect between reasoning and planning, as well as the existence of shortcut learning dependent on text priors.

decomposed CoT subtasks. Since there is no explicit reasoning process, we mainly examine the attention distribution between the sequence of image tokens and the sequence of textual prior tokens in the input across different depth layers during model output.

The results in Table 8 demonstrate that *Omnidrive* progressively focuses more on the image sequence during detection tasks (implicit reasoning). At the final layer, its attention to image tokens reaches 15.92%, while attention to the textual prior sequence fluctuates stably between approximately 6-7%, significantly lower than that for image tokens. This indicates that the textual prior component does not play a decisive role in the detection task that provides the implicit reasoning ability.

During planning tasks, *Omnidrive* consistently allocates significantly high attention weights to textual prior sequences across different layers, ultimately reaching approximately 24%, which is much higher than the attention weights to the image sequence. This demonstrates that *Omnidrive* also heavily relies on textual prior information for planning. The attention weights assigned to the image sequence by *Omnidrive* in the planning task also increase slightly as the layer depth increases. This may stem from the absence of preceding reasoning sequences compared to *CoT_grpo*, resulting in fewer tokens being processed, and consequently increasing attention to images. It may also reflect the model's implicit, superficial and limited processing of image information, as the final attention weight only accounts for 4%.

The results of the sequence-level attention analysis on *Omnidrive* further support our conclusion: current VLM driving agents heavily rely on textual priors during planning, with reasoning processes being almost irrelevant. This also shows that counterfactual reasoning fails to resolve this issue. Furthermore, results also show that modality bias in comprehension persists within *Omnidrive*.

## A.9 MORE EXAMPLES OF CAUSAL PROBE

This section provides more examples of our causal probe applied to diverse driving scenarios, further illustrating the reasoning-planning disconnect and the efficacy of our diagnostic method.

Figure 3 illustrates the application of the causal probe in a simple straight driving scenario. In the baseline case with no perturbation, both the *CoT_grpo* and *Omnidrive* agents correctly produce a plan to continue straight. However, when a minor lateral offset is introduced into the textual priors, both agents exhibit catastrophic planning failures. Their planned trajectories deviate significantly to the left or right, directly following the perturbed history. Crucially, for the *CoT_grpo* agent, the generated reasoning remains correct and consistent across all three cases (e.g., "Reasoning: continue accelerating and minor steer"). Despite the correct reasoning, the planning module produces a wildly

divergent and unsafe trajectory. This provides a clear, qualitative example of the planning module ignoring the reasoning module and instead relying on the shortcut provided by the textual priors.

Figure 4 provides an even more stark example of disconnect in a stationary scenario. In the baseline case, both agents correctly plan to remain stopped at a red light while pedestrians are present. Their reasoning correctly identifies the red light and the need to wait. However, when a lateral offset is applied to the velocity priors, both agents generate an unsafe plan that involves a significant and unnecessary lateral swerve, even while stationary. This case is particularly revealing. The reasoning module for *CoT_grpo* continues to correctly state that the agent should remain stopped due to the red light and pedestrians. Yet, the planning module, completely decoupled from this reasoning, still reacts to the perturbed prior and outputs an erroneous, non-zero motion plan. This demonstrates how deeply ingrained the shortcut learning is, as it can override even a fundamental safety behavior like remaining stopped at a red light.

### A.10 EVALUATION BY SCENARIO COMPLEXITY

To validate the consistency of our findings across different scene complexities, we stratify the 14 distinct evaluation types in the nuPlan challenge test set to the hard, general and simple categories, as stated in Appendix A.3. We measure the performance of the standard agent (*CoT*), the blind agent (*Plan_NoV*), and the prior-deprived agent (*CoT_NoPri*) across these categories. The evaluation results of the 3s planning are summarized in Table 9.

As observed in the table, the "blind" *Plan_NoV* agent achieves parity with (or slightly exceeds) the multimodal *CoT* agent across all complexity levels, including "Hard" scenarios. This confirms that the reasoning-planning disconnect is not limited to simple cases; even in complex turning scenarios, the model ignores visual cues in favor of priors. The performance gap between the standard agent and the prior-removed agent ($\Delta$ (*CoT* - *CoT_NoPri*)) serves as a proxy for "reliance on priors." Notably, this gap widens significantly as complexity increases: the Open-Loop score of *CoT_NoPri* drops by 47.16 points in Hard scenarios compared to only 18.50 points in Simple scenarios. These results confirm that the model's reliance on shortcuts is not only consistent but also effectively strengthened under challenging conditions. Rather than engaging in deeper visual reasoning for complex tasks, the agent leans more heavily on textual priors to solve the problem.

### A.11 VISUAL INPUT PERTURBATION

In the main text, we use our proposed causal probes to perturb the textual prior and validate the reasoning-planning decoupling in the VLM-driving agent and the existing shortcut learning from text priors. To further strengthen our conclusions, we perturb the visual inputs to the CoT model during the testing phase in two additional ways:

- Scene Replacement (*CoT_V_Replace*): Randomly replacing the correct visual scene with a contradictory one.
- Severe Noise (*CoT_V_Crop_Noise*): Randomly crop the input image and add Gaussian noise.

Table 10 reports the planning performance of CoT under two forms of visual perturbation, showing that such perturbations exert negligible influence on the resulting plans. Together with the results from the *Plan_NoV*, which removes all visual input and thus represents the most extreme perturbation, these results collectively demonstrate that the planning ability of the trained VLM driving agent is effectively visually inert; it functions by truly relying on textual shortcuts.

Table 9: Open-loop and closed-loop performance comparison across scenario categories.

| Category | Scenario | Open-Loop Score ↑ | | | Closed-Loop Score ↑ | | |
|---|---|---|---|---|---|---|---|
| | | CoT | CoT_NoPri | Plan_NoV | CoT | CoT_NoPri | Plan_NoV |
| Hard | high_lateral_acceleration | 89.69 | 40.74 | 90.04 | 90.31 | 66.18 | 90.51 |
| | behind_long_vehicle | 98.31 | 67.25 | 98.57 | 99.28 | 77.74 | 99.45 |
| | changing_lane | 92.33 | 27.48 | 92.40 | 85.96 | 67.45 | 86.08 |
| | starting_left_turn | 86.98 | 36.94 | 87.26 | 82.93 | 69.91 | 83.41 |
| | starting_right_turn | 88.64 | 42.56 | 88.94 | 84.63 | 68.14 | 85.49 |
| | near_multiple_vehicles | 93.42 | 42.72 | 93.51 | 92.79 | 81.82 | 92.67 |
| | traversing_pickup_dropoff | 89.80 | 51.35 | 89.99 | 84.03 | 69.27 | 84.26 |
| **Hard Avg** | | **91.31** | **44.15** | **91.53** | **88.56** | **71.50** | **88.84** |
| Δ (CoT - CoT_NoPri) | | | **+47.16** | | | **+17.06** | |
| General | waiting_for_pedestrian_to_cross | 90.14 | 44.68 | 90.81 | 95.27 | 82.76 | 95.06 |
| | starting_straight_traffic_light_intersection_traversal | 91.84 | 42.78 | 92.09 | 93.57 | 81.49 | 93.26 |
| | stopping_with_lead | 93.03 | 67.84 | 94.87 | 94.14 | 81.75 | 94.35 |
| | following_lane_with_lead | 95.16 | 67.07 | 96.14 | 99.98 | 91.49 | 99.98 |
| **Gen. Avg** | | **92.54** | **55.59** | **93.48** | **95.74** | **84.37** | **95.66** |
| Δ (CoT - CoT_NoPri) | | | **+36.95** | | | **+11.37** | |
| Simple | low_magnitude_speed | 92.71 | 68.73 | 92.61 | 95.26 | 85.24 | 95.37 |
| | stationary_in_traffic | 99.29 | 94.47 | 99.08 | 99.06 | 97.74 | 98.93 |
| | high_magnitude_speed | 94.55 | 67.86 | 94.56 | 96.13 | 87.72 | 96.78 |
| **Simple Avg** | | **95.52** | **77.02** | **95.42** | **96.82** | **90.23** | **97.03** |
| Δ (CoT - CoT_NoPri) | | | **+18.50** | | | **+6.58** | |

Table 10: Evaluation of the *CoT*'s planning performance (3-second horizon) under replacement and crop_noise visual input perturbations.

| Agent | Open-Loop Score ↑ | Close-Loop Score ↑ |
|---|---|---|
| CoT | 92.56 | 92.38 |
| CoT_V_Crop_Noise | 91.86 | 90.79 |
| CoT_V_Replace | 92.36 | 92.27 |

**Prompt for Traffic Light Reasoning:**

 <If the light stays the same, what can ego-vehicle do? If it changes, what can ego-vehicle do? Take into account the surrounding environment and the ego-vehicle's current state.>

**Prompt for Traffic Sign Reasoning:**

<Identify any traffic signs visible in the front views, describe their meaning, and explain whether they affect the current movement of the ego-vehicle. If no signs are detected, state that no traffic signs are visible ahead.>

 **Prompt for Weather Reasoning:**

<Give a short description of whether in the current driving scenario, then briefly analyze if the weather poses a threat to the ego-vehicle's current driving, and if it poses a threat, what should the ego-vehicle do.>

**Prompt for Detected Vehicle Reasoning:**

<Briefly analyze the vehicle's intent and whether the vehicle may threaten the driving of the ego-vehicle, and if it poses a threat, analyze what kind of threat it would be. Your analysis should refer to the 'Addition info' but should not include the 'Addition info'. 'Addition info: The position of the object in the past 1 second is ({x_past:.2f},{y_past:.2f}).>

**Prompt for Detected Pedestrian Reasoning:**

<Briefly analyze the motion of the pedestrian, then analyze whether the pedestrian may threaten the driving of the ego-vehicle, and if it poses a threat, analyze what kind of threat it would be. Your analysis should refer to the 'Addition info' but should not include the 'Addition info'. Addition info: The position of the object in the past 1 second is ({x_past:.2f},{y_past:.2f})>

**Prompt for Detected Bicycle Reasoning:**

<Briefly analyze the bicycle's intent and whether it may threaten the driving of the ego-vehicle, and if it poses a threat, analyze what kind of threat it would be. Your analysis should refer to the 'Addition info' but should not include the 'Addition info'. Additional info: The position of the object in the past 1 second is ({x_past:.2f},{y_past:.2f}).>

 **Prompt for Driving Decision Reasoning:**

<Give a brief driving decision based on all the current information and analysis, and common sense, traffic principles in driving. It should clearly guide the ego-vehicle's action over the next 3 seconds. You should specify the anticipated maneuver, provide the anticipated changes in ego-vehicle's position (x, y) and velocity (vx, vy) according to the maneuver in the near future 3 seconds and the reasons.

While the future trajectory is known, your tone should remain predictive: extrapolating logically from the present and can not report the known future data in your driving decision. Use inferential language, but do NOT use non-predictive language like: according to the future. Your decision must be grounded in present state logic and reflect the causal relationships between the current environment and the driving decision. Remember to remain predictive when extrapolating.>

Figure 5: Prompt for GPT-4.1 to generate analysis for different parts. All together combine to form the plan-aligned CoT in DriveMind.

**Input Multi-view Images:**

**Input Text Prompt:**

Here are three stitched images representing the current driving scene, captured from eight cameras mounted on the ego-vehicle that you are driving:
-Image 1 (left to right): Front-left (CAM_L0), Front (CAM_F0), Front-right (CAM_R0)
-Image 2 (left to right): Left (CAM_L1), Right (CAM_R1)
-Image 3 (left to right): Rear-right (CAM_R2), Rear (CAM_B0), Rear-left (CAM_L2)
The ego-vehicle your are driving has the dimension (length, width and height): (5.18,2.30,1.78).
**Current ego-vehicle state: location (0.0, 0.0), velocity (3.71,-0.10), total velocity magnitude 3.72.**
Lane speed limit: 3.24m/s.
**Navigation: Center point of the next target road block: (29.58, 10.62).**
**Past 3 seconds ego-vehicle states:**
**- at past 3.0 s: location (-10.39, 1.22), velocity (3.21, -1.27), total velocity magnitude 3.45...**
**- at past 2.0 s: location (-7.14, 0.45), velocity (3.37, -0.85), total velocity magnitude 3.47...**
**- at past 1.0 s: location (-3.66, 0.06), velocity (3.75, -0.45), total velocity magnitude 3.78...**
Your task is to give the trajectory and speed of the ego-vehicle for the next 3 seconds at 0.2 second prediction intervals based on the current camera data, known information about the ego-vehicle, and navigation information. Please analyze the driving environment first, you should consider aspects as follows step by step: Lane Analysis, Traffic Light Analysis, Traffic Sign detection ahead the ego-vehicle, Weather Analysis, Detected critical Vehicles, Detected critical Pedestrians, Detected critical bicycles, Detected other objects. Then, based on all the information, combined with Common Sense in driving, Traffic Principles and reasonable assumptions about the future, give a macro driving decision that decides how the ego-vehicle should act over the next 3 seconds. Finally, give the predicted ego-vehicle states—location (x, y) and velocity (vx, vy)—over the next 3.0 seconds. All of your analysis including the macro driving decision should be enclosed in a pair of <think></think>, and your final predictions should be enclosed in a pair of <answer></answer>.

---

**Ground Truth plan-aligned CoT Answer:**

**<think>**
Now I'm driving in the lane.
**Lane Analysis:**
2 lanes detected from ego-vehicle's cameras. Taking the heading direction of the ego-vehicle and the location of the ego-vehicle as references, lanes in order from left to right are:
- Lane on the left side, same direction as the ego-vehicle's heading. Lane width: 4.35.
- Lane the ego-vihicle in, same direction as the ego-vehicle's heading. Lane width: 4.95.
**Traffic Light Analysis:**
No traffic lights detected in the current scenario.
**Traffic Sign detection ahead the ego-vehicle:**
No traffic signs are visible ahead.
**Weather Analysis:**
It is daytime with clear weather, as evidenced by bright sunlight, clear shadows, and good visibility in the images. The weather does not pose any threat to the ego-vehicle's driving.
**Detected critical Vehicles:**
-a silver sedan with a black roof and a silver logo on the front hood, can be seen from ['CAM_R0', 'CAM_R1'], now at location (2.62,-6.57), at the front-right of the ego vehicle, dimension (4.92,1.87,1.53). The vehicle seems moving slowly forward and very slightly to the right. It is likely waiting, parked, or maneuvering in a drop-off area and does not currently threaten the ego-vehicle, provided the ego-vehicle remains in its lane.
-a black hatchback with a logo on the front hood, can be seen from ['CAM_R1', 'CAM_R2'], now at location (-5.30,-6.74), at the rear-right of the ego vehicle, dimension (4.37,1.83,1.51). The vehicle seems slowly moving forward and slightly to the right. It seems to be maneuvering through a congested drop-off area, and is not actively threatening the ego-vehicle as it stays well behind and to the right.
-an orange pickup truck with large tires, can be seen from ['CAM_B0', 'CAM_L1', 'CAM_L2'], now at location (-8.34,5.92), at the rear-left of the ego vehicle, dimension (6.45,2.24,2.20). The vehicle is moving very slowly forward and slightly left. Its position suggest it is unlikely to enter the ego-vehicle's path, so it does not pose a current threat.
**Detected critical Pedestrians:**
-a pedestrian, can be seen from ['CAM_B0', 'CAM_L2'], now at location (-16.40,8.26), at the rear-left of the ego vehicle, dimension (1.04,0.87,1.92). The pedestrian is moving forward and significantly toward the left, but is located far behind the ego-vehicle and in a separate area, presenting no threat.
**Detected critical bicycles:**
- No bicycle detected.
**Detected other objects:**
- an object, at location (1.58,1.96), at the front-left of the ego vehicle, dimension (0.44,0.45,1.22).
- an object, at location (-5.95,2.72), at the rear-left of the ego vehicle, dimension (0.43,0.44,1.23).
**Macro driving decision:**
The ego-vehicle is currently traveling at a speed slightly above the lane speed limit, progressing in a gentle leftward curve as seen in the current road geometry. There are no obstacles or active threats detected in the immediate path, with all identified vehicles and pedestrians remaining clear of the driving lane. The environment is controlled and slow-paced, typical of a hotel drop-off or pickup loop with some stationary or slow vehicles and scattered pedestrians. Over the next 3 seconds, the ego-vehicle should anticipate a continued gentle left curve to remain centered in the lane while gradually adjusting speed closer to the posted limit to ensure both compliance and safety. The vehicle is expected to shift incrementally leftward (increasing positive y), as demonstrated by recent velocity changes, while longitudinal velocity (vx) gently decreases as it adapts for the curve and aligns with lane limits. This maneuver promotes lane-keeping, smoothness, and readiness for any evolving activity in the area ahead. The vehicle should stay alert for possible merging or entering traffic near the curbside but does not need to take evasive action given the current clear path.
**</think>**

**<answer>**
**Based on current environmental perception with comprehensive analysis, driving decision, and reasonable assumptions about the future, the predicted ego-vehicle states—location (x, y) and velocity (vx, vy)—over the next 3.0 seconds are as follows:**\n- at future 0.2 s: location (0.72, 0.04), velocity (3.69, 0.01), total velocity magnitude 3.69.\n- at future 0.4 s: location (1.44, 0.09), velocity (3.65, 0.07), total velocity magnitude 3.65.\n- at future 0.6 s: location (2.15, 0.15), velocity (3.62, 0.13), total velocity magnitude 3.63.\n- at future 0.8 s: location (2.84, 0.22), velocity (3.58, 0.21), total velocity magnitude 3.59.\n- at future 1.0 s: location (3.53, 0.32), velocity (3.54, 0.28), total velocity magnitude 3.55.\n- at future 1.2 s: location (4.21, 0.43), velocity (3.50, 0.35), total velocity magnitude 3.52.\n- at future 1.4 s: location (4.87, 0.55), velocity (3.45, 0.42), total velocity magnitude 3.48.\n- at future 1.6 s: location (5.52, 0.68), velocity (3.40, 0.51), total velocity magnitude 3.43.\n- at future 1.8 s: location (6.16, 0.83), velocity (3.35, 0.56), total velocity magnitude 3.39.\n- at future 2.0 s: location (6.79, 0.98), velocity (3.30, 0.63), total velocity magnitude 3.36.\n- at future 2.2 s: location (7.42, 1.15), velocity (3.26, 0.67), total velocity magnitude 3.33.\n- at future 2.4 s: location (8.03, 1.32), velocity (3.21, 0.73), total velocity magnitude 3.29.\n- at future 2.6 s: location (8.64, 1.50), velocity (3.16, 0.79), total velocity magnitude 3.26.\n- at future 2.8 s: location (9.24, 1.69), velocity (3.12, 0.82), total velocity magnitude 3.23.\n- at future 3.0 s: location (9.82, 1.89), velocity (3.09, 0.87), total velocity magnitude 3.21.
**</answer>**

Figure 6: An example of our plan-aligned CoT driving VQA data from DriveMind, with the three priors in the text input marked as green.

