# OpenReview forum: "More Than Meets the Eye? Uncovering the Reasoning-Planning Disconnect in Training Vision-Language Driving Models"
_ICLR.cc/2026/Conference — Submitted to ICLR 2026_

### Official Review · Reviewer_xNnf · 2025-10-29

**Soundness:** 3
**Presentation:** 3
**Contribution:** 2
**Rating:** 6
**Confidence:** 4

**Summary:**

This paper investigates the causal link between reasoning and planning in VLM driving agents. It introduces the DriveMind dataset and a "Causal Probe" to test this. The findings suggest a "reasoning-planning disconnect," where agents use textual priors as shortcuts and reasoning serves as a non-causal byproduct.

**Strengths:**

This paper has clear significance, as it addresses the important assumption of causality in reasoning-based VLM agents. The quality of the work is solid, highlighted by the creation of the DriveMind dataset for causal analysis and a structured experimental plan . The "Causal Probe" is a novel contribution for diagnostics. The paper is clearly written, and the "Reasoning-Planning Decoupling Hypothesis" is supported by the ablation and attention analyses. This work raises valuable questions about the nature of interpretability in these models.

**Weaknesses:**

The paper's setup for CoT generation and training is worth discussing. The CoT is generated by GPT-4.1 from ground truth and then used as a supervision target alongside the plan (Input $\rightarrow$ [CoT, Plan]). This joint SFT objective might allow the model to learn two separate tasks, both of which could be predicted from textual priors. It would be helpful for the authors to discuss if this disconnect might be an artifact of this joint training. Exploring a sequential setup (e.g., Input $\rightarrow$ CoT, then [Input, Model's_Own_CoT] $\rightarrow$ Plan) could provide further insight.

**Questions:**

1. Could the joint SFT objective for both CoT and plan tokens contribute to the observed disconnect? I am curious if the authors considered or tested a sequential training setup (e.g., training a planning module on the model's own generated reasoning) and whether that might change the causal link.
2. Regarding the "Causal Probe" results in Figure 2 , how frequently was the direct contradiction (where the CoT's stated logic mismatches the plan's action) observed across the full test set?
3. The GRPO results showing reinforced shortcutting are interesting. Could the authors elaborate on the choice of the "format" reward and whether they think a different reward (perhaps one that scores the alignment of the CoT with visual evidence) might lead to a different outcome?

---

> ### Author Response · Authors · 2025-11-21
> **Author Response (1)**
>
> We sincerely appreciate the reviewer's thorough review of our paper and the positive feedback. We are grateful for the time and effort the reviewer dedicated to providing constructive comments. We would like to address all of the points raised in the review and provide additional context or clarification that we believe could contribute to an even more favorable evaluation.
>
> > W1: The paper's setup for CoT generation and training is worth discussing. The CoT is generated by GPT-4.1 from ground truth and then used as a supervision target alongside the plan (Input $\rightarrow$ [CoT, Plan]). This joint SFT objective might allow the model to learn two separate tasks, both of which could be predicted from textual priors. It would be helpful for the authors to discuss if this disconnect might be an artifact of this joint training. Exploring a sequential setup (e.g., Input $\rightarrow$ CoT, then [Input, Model's_Own_CoT] $\rightarrow$ Plan) could provide further insight.
> > Q1: Could the joint SFT objective for both CoT and plan tokens contribute to the observed disconnect? I am curious if the authors considered or tested a sequential training setup (e.g., training a planning module on the model's own generated reasoning) and whether that might change the causal link.
>
> **A1:**
> We appreciate this highly insightful observation regarding the joint SFT objective. We argue that the disconnect is fundamentally robust to the sequential structure of the training process.
>
> Our baseline findings have already demonstrated that the disconnect is not dependent on the joint CoT task itself. The baseline Plan agent (which is trained only to produce Plan tokens, effectively decoupling the tasks entirely) achieves planning scores nearly identical to the joint SFT CoT agent. This demonstrates that the high performance and reliance on priors are inherent to the planning part itself.
>
> To directly address your suggestion, we conduct a two-stage sequential training experiment, training agents (CoT\_Seq and CoT\_Seq\_NoPri) using the setup Input $\rightarrow$ CoT, followed by [Input, Model's_Own_CoT] $\rightarrow$ Plan.
>
> | Agent | Open-Loop Score ↑| Close-Loop Score ↑|
> | :--- | :---: | :---: |
> | CoT_Seq | 91.64 | 91.45 |
> | CoT_Seq_NoPri | 49.61 | 74.62 |
>
> The performance patterns of the sequential agents are similar to the joint SFT setting. The CoT_Seq agent achieves high scores, but its planning ability drops drastically when priors are removed (CoT_Seq_NoPri), confirming that this two-stage training approach still fails to break the reliance on textual priors. The disconnect is thus fundamental to the architectural vulnerability, regardless of whether the tasks are learned jointly or sequentially. We acknowledge that exploring alternative sequential training methods is a highly insightful thinking direction. We view this as orthogonal to our current work and an important path for future research aimed at breaking the reasoning-planning disconnect.

---

> ### Author Response · Authors · 2025-11-21
> **Author Response (2)**
>
> >Q2: Regarding the "Causal Probe" results in Figure 2 , how frequently was the direct contradiction (where the CoT's stated logic mismatches the plan's action) observed across the full test set?
>
> **A2:** The quantitative evidence confirms that the direct contradiction (where the agent's $\text{CoT}$ logic mismatches its final plan) is a systematic failure mode that is triggered across all tested scenarios at minimal levels of textual perturbation, strongly supporting the Reasoning-Planning Decoupling Hypothesis. The consistency of this failure is demonstrated by the results of the two Causal Probe methods:
>
> - Lateral Direction Inversion: This probe tests scenarios where the visual scene (e.g., a left curve) contradicts the falsified historical movement (e.g., a simulated drift to the right). The planning output directionally flips to follow the falsified prior 100% of the time for the samples tested on CoT\_grpo and Omnidrive ($\text{Inversion Rate} = 100\\\%$). Manual inspection confirms that in all of these instances, the agent's generated CoT correctly articulate the reasoning based on the visual scene, but the final plan execute the inverse action. We have addressed this on page 9, line 480 of the revised version.
>
> - Lateral Offset Perturbation: This probe tests stability by introducing a small lateral offset ($\delta$) to the textual prior. The LLM agents exhibit an immediate collapse in stability as the perturbation magnitude increase. To better quantify the model's sensitivity, we extend this probe to report both the Mean Final Deviation (MFD) (m) of the trajectory endpoint and the Relative Intervention Degree (RID). The MFD is defined as:$$MFD = \frac{\sum_{i=1}^{n} |Original\\\_Final\\\_xy_{i} - Perturbed\\\_Final\\\_xy_{i}| }{n},$$
>  where n is the number of our tested samples.
> RID is defined as the ratio of the difference between a sample's output before and after perturbation to the theoretical error that would occur if the model perfectly followed the perturbation (i.e., $\delta \cdot v_i \cdot T$):$$RID =\frac{1}{n}\sum_{i=1}^{n}\frac{ |Original\\\_Final\\\_xy_{i} - Perturbed\\\_Final\\\_xy_{i}|}{\delta \cdot v_i \cdot T},$$ where $v_{i}$ is the ego-velocity of the i-th sample and $T$ is the prediction horizon. This normalized comparison enables RID to quantitatively measure the extent to which a model relies on textual priors: RID=1 indicates complete reliance on the error in textual priors, RID>1 indicates amplification of the error, and RID<1 indicates the ability to correct the error through visual reasoning. This provides a scale-independent, robust metric that allows for cross-scenario comparisons.
>
>     Our results are as below:
>
>     | $\delta$       | 0.02  | 0.06  | 0.08  | 0.10  | 0.30   |
>     |----------------|-------|-------|-------|-------|--------|
>     | MFD ↓ | 0.55 | 2.17 | 3.75 | 5.89 | 15.46 |
>     | RID ↓| 1.00  | 1.47  | 2.04  | 2.17  | 2.04   |
>     | Unaffected Ratio | 39.3%  | 0  | 0  | 0 | 0   |
>
>     While approximately $39.3\\\%$ of samples were initially unaffected (final deviation < 0.1) at the lowest level ($\delta=0.02$), all samples (100%) are disturbed by the time the perturbation reached $\delta \ge 0.06$. Crucially, during these disturbances, the generated CoT remained correct. The subsequent planning failure (lateral swerve or deviation)  demonstrates that the planning module systematically ignores its own reasoning to follow the noisy textual input.

---

> ### Author Response · Authors · 2025-11-21
> **Author Response (3)**
>
> > Q3: The GRPO results showing reinforced shortcutting are interesting. Could the authors elaborate on the choice of the "format" reward and whether they think a different reward (perhaps one that scores the alignment of the CoT with visual evidence) might lead to a different outcome?
>
> **A3:** The "format" reward is chosen to impose structural discipline on the reasoning process: It forces the model to engage the CoT module and adhere to the required structural sequence ($\langle \text{think} \rangle \rightarrow \langle \text{answer} \rangle$). It implicitly encourages the model to analyze the scene from input to output, upholding analytical causality. It standardizes the output format, which is necessary for trajectory extraction and supervision.
>
>
> To test whether forcing the model to connect the CoT to visual evidence is needed,  we implement a new reward scoring the alignment of the CoT with key visual components:
> $$  reward\\\_visual\\\_match = \frac{count(key\\\_ visual\\\_components\\\_in\\\_CoT)}{count(all\\\_key\\\_visual\\\_components)},$$
> where "key visual components" are concretely defined as the critical environmental elements extracted from the ground truth that directly influence driving decisions, specifically: active traffic lights, relevant road signs, and dynamic obstacles such as pedestrians or vehicles within a safety radius. The reward is computed as the ratio of these ground-truth elements explicitly mentioned in the agent's generated CoT to the total number of such elements actually present in the scene.
>
>
>
> We have trained Base\_grpo\_visual\_match agent and CoT_grpo_visual_match using this objective, both achieving high average matching scores greater than $0.7$. However, under our causal probe ($\delta=0.1$), these driving agents still exhibite large deviation compared to the unperturbed baseline:
> | Agent | MFD ↓ | RID ↓ |
> | :--- | :---: | :---: |
> | CoT_grpo_visual_match | 5.77 | 1.99 |
> | Base_grpo_visual_match | 4.42 | 1.61 |
>
> This catastrophic failure demonstrates that even when the model is successfully incentivized to perceive and articulate the correct visual facts (e.g., correctly identifying a red light in text), the planning module still bypasses this reasoning to rely on textual priors. The issue is thus a fundamental architectural disconnect rather than a lack of visual attention supervision. Our future work will specifically focus on developing and exploring such visual causality rewards to rebuild the reasoning-planning link, aligning with the discussions in Section 5.
>
> The reviews provided by this reviewer offer valuable insights that will meaningfully guide our future work. We once again express our sincere gratitude for the reviewer’s thorough evaluation, constructive questions, and supportive comments. Should the reviewer have any further questions, we would be delighted to continue the discussion at any time.

---

### Official Review · Reviewer_Head · 2025-11-01

**Soundness:** 3
**Presentation:** 3
**Contribution:** 2
**Rating:** 4
**Confidence:** 3

**Summary:**

This paper studies a key assumption in recent VLM-based driving agents: whether CoT reasoning meaningfully contributes to downstream planning decisions. To address this question, the authors introduce DriveMind, a dataset comprising approximately 50K VQA-style samples derived from nuPlan, with GPT-4.1-generated CoT annotations and verified through human validation. The authors fine-tune three VLM models on DriveMind using both SFT and GRPO, conducting systematic ablations that remove CoT, visual inputs, or textual priors.

The empirical findings reveal a interesting pattern: planning agents without visual information perform comparably to their vision-enabled counterparts, while removing textual priors significantly degrades performance. This suggests that existing VLM driving agents may exhibit shortcut learning by over-relying on textual priors rather than engaging in genuine visual reasoning.

To mechanistically probe this phenomenon, the paper introduces a sequence-level attention analysis demonstrating that during planning, attention mass concentrates predominantly on textual priors rather than on preceding CoT tokens or image features. Additionally, the authors propose a training-free causal probe to evaluate model robustness under perturbed prior conditions.

**Strengths:**

1. well‑motivated question with a clear negative result.: The paper addresses a fundamental assumption in VLM-based driving: whether CoT reasoning causally contributes to planning. The ablation results are striking: vision-deprived agents (Plan_NoV) achieve performance comparable to full vision+CoT agents on nuPlan metrics, while removing textual priors causes catastrophic degradation. This demonstrates a clear reasoning-planning disconnect and represents an important cautionary finding for the field.

2. Rigorous diagnostics: The paper uses complementary diagnostic approaches: (a) sequence-level attention analysis quantitatively shows that planning attends primarily to textual priors rather than CoT or images, while reasoning generation shows appropriate visual attention; (b) training-free causal probes via perturbations qualitatively expose agent brittleness to prior modifications, revealing contradictions between stated reasoning and actual planning. This multi-faceted evidence strengthens the central claims.

3. Reproducibility: All the training settings and nuPlan metric definitions are carefully documented in Appendix, which is valuable for replication and for the communities to do follow-up analysis.

**Weaknesses:**

1. Limited contributions: While the paper compellingly demonstrates the reasoning-planning disconnect through attention analysis and perturbation studies, it does not consolidate these insights into a standardized, reusable evaluation metric or benchmark. The causal probe remains qualitative (e.g., visual inspection of trajectory contradictions) rather than providing a quantitative robustness score that could be systematically computed and compared across models.

2. Insufficient justification of dataset novelty and quality: The paper positions DriveMind as addressing semantic gaps in nuScenes-based datasets and sim-to-real gaps in CARLA-based datasets, but lacks direct comparative analysis. Specifically: (a) missing comparison table showing how DriveMind differs from existing datasets (DriveLM, DriveVLM, DriveCoT, OmniDrive) in terms of semantic richness, CoT quality, or modular structure; (b) Potential data leakage: CoT generation explicitly provides GPT-4.1 with future expert trajectories, which may introduce subtle leakage despite stylistic constraints; (c) Missing ablation on dataset quality: It's unclear whether the observed disconnect is specific due to DriveMind's distributional bias and data quality, or would generalize to other datasets.

3. Limited experimental scope and generalizability: The primary experiments focus on 7B-parameter models (Qwen2.5-VL, LLaVA-1.6) and an OmniDrive reimplementation. This raises several concerns: (a) Model scale: The findings may not generalize to larger VLMs (like 32B+), which may exhibit different reasoning capabilities and attention patterns. Testing at least one larger model would strengthen generalizability claims; (b) Proprietary models: The absence of closed-source model evaluation (e.g., GPT-4o as a driving agent) makes it unsure whether this is an artifact of open-source VLM training or a broader issue.

**Questions:**

1. Is the reasoning-planning disconnect specific to this model scale and Drivemind dataset, or does it generalize to larger models and other training recipes? Can the authors test larger open-source models or evaluate closed-source models in zero-shot settings to demonstrate broader applicability?

2. Given that textual priors enable shortcut learning, did the authors experiment with alternative input designs or tasks that reduce prior specificity—such as coarse-grained navigation, noisy ego states, or prompts that require explicit visual grounding? This would clarify whether the issue is fundamental or can be mitigated through careful task design.

3. The causal probe perturbs textual priors. Have the authors tried perturbing visual inputs (e.g., masking, noise, scene replacement)? If planning truly relies on textual shortcuts, visual perturbations should have minimal impact.

---

> ### Author Response · Authors · 2025-11-21
> **Author Response (1)**
>
> We sincerely appreciate the reviewer’s time and thoughtful evaluation of our paper. We hope that our responses adequately address all of the concerns. We have also uploaded a revised version of the manuscript with the corresponding updates, and we kindly invite the reviewer to refer to it alongside our point-by-point replies:
>
> > W1: Limited contributions: While the paper compellingly demonstrates the reasoning-planning disconnect through attention analysis and perturbation studies, it does not consolidate these insights into a standardized, reusable evaluation metric or benchmark. The causal probe remains qualitative (e.g., visual inspection of trajectory contradictions) rather than providing a quantitative robustness score that could be systematically computed and compared across models.
>
> **A1:** We clarify that our core contribution lies in the discovery and rigorous diagnosis of the Reasoning-Planning Decoupling Hypothesis. To solidify this contribution and move beyond qualitative observations, we concur that a systematically computable metric is essential. We address this by transforming our Causal Probe methodology into a standardized quantitative robustness score that can be systematically computed and compared across models. To better quantify the model's sensitivity, we report the Mean Final Deviation (MFD) (m) of the trajectory endpoint and the Relative Intervention Degree (RID). The MFD is defined as:$$MFD = \frac{\sum_{i=1}^{n} |Original\\\_Final\\\_xy_{i} - Perturbed\\\_Final\\\_xy_{i}| }{n},$$ where n is the number of the test samples.
>
> RID is defined as the ratio of the difference between a sample's output before and after perturbation to the theoretical error that would occur if the model perfectly followed the perturbation (i.e., $\delta \cdot v_i \cdot T$):$$RID =\frac{1}{n}\sum_{i=1}^{n}\frac{ |Original\\\_Final\\\_xy_{i} - Perturbed\\\_Final\\\_xy_{i}|}{\delta \cdot v_i \cdot T},$$ where $v_{i}$ is the ego-velocity of the i-th sample and $T$ is the prediction horizon. This normalized comparison enables RID to quantitatively measure the extent to which a model relies on textual priors: RID=1 indicates complete reliance on the error in textual priors, RID>1 indicates amplification of the error, and RID<1 indicates the ability to correct the error through visual reasoning. This provides a scale-independent, robust metric that allows for cross-scenario comparisons.
>
> We expand our analysis by conducting a sensitivity analysis on the CoT_grpo agent. We utilize a test set of 300 randomly sampled scenarios and vary the perturbation magnitude $\delta$ from 0.02 to 0.3:
> | $\delta$       | 0.02  | 0.06  | 0.08  | 0.10  | 0.30   |
> |----------------|-------|-------|-------|-------|--------|
> | MFD  ↓          | 0.55 | 2.17 | 3.75 | 5.89 | 15.46 |
> | RID ↓  | 1.00  | 1.47  | 2.04  | 2.17  | 2.04   |
>
> This framework provides a standardized evaluation protocol because it relies on fixed methodologies and mathematically defined metrics rather than subjective assessment. By pairing the DriveMind dataset (which provides the modular structure necessary for intervention) with this Causal Probe, we create a **reusable benchmark**. We have included our quantitative metrics on line 332 of page 7 in the revised version and updated the results of this probe experiment on line 465 of page 9.

---

> ### Author Response · Authors · 2025-11-21
> **Author Response (2)**
>
> > W2: Insufficient justification of dataset novelty and quality: The paper positions DriveMind as addressing semantic gaps in nuScenes-based datasets and sim-to-real gaps in CARLA-based datasets, but lacks direct comparative analysis. Specifically: (a) missing comparison table showing how DriveMind differs from existing datasets (DriveLM, DriveVLM, DriveCoT, OmniDrive) in terms of semantic richness, CoT quality, or modular structure; (b) Potential data leakage: CoT generation explicitly provides GPT-4.1 with future expert trajectories, which may introduce subtle leakage despite stylistic constraints; (c) Missing ablation on dataset quality: It's unclear whether the observed disconnect is specific due to DriveMind's distributional bias and data quality, or would generalize to other datasets.
>
> **A2:**
> - (a) Comparison Table: We have included a detailed table (as shown below) at the top of Page 5 in the revision comparing DriveMind against prominent multimodal driving datasets (DriveLM, DriveVLM, DriveCoT, OmniDrive) across key dimensions. This will explicitly highlight DriveMind's unique combination of rich, structured semantic context from real-world logs and its modular structure specifically designed for causal ablation.
>
> | Dataset | Real-World Logs | Semantic Richness | Expert CoT | Modular Structure | Scenario Coverage |
> | :--- | :--- | :--- | :--- | :--- | :--- |
> | DriveLM | Mixed | Moderate (nuScenes), High (CARLA) | Yes | No | --- |
> | DriveVLM | Yes | Moderate (SUP-AD) | Yes | No | 40 |
> | DriveCoT | No | High (CARLA) | Yes | No | 5 |
> | OmniDrive | Yes | Moderate (nuScenes) | No | No | --- |
> | **DriveMind (Ours)** | **Yes** | **High (nuPlan)** | **Yes** | **Yes** | **61** |
>
>
> - (b) Potential Data Leakage: We respectfully clarify the GPT-4.1 process. The expert trajectory is provided to GPT-4.1 as context to ensure the generated reasoning (the causal logic) is factually correct, not to generate the trajectory itself. We deliberately imposed constraints (Appendix A.2, Figure 5), such as "Use inferential language, but do NOT use non-predictive language like: according to the future," to ensure the language of the CoT remains predictive and causal, thereby guaranteeing no leakage from the ground-truth planning. We additionally verified adherence through human inspection.

---

> ### Author Response · Authors · 2025-11-21
> **Author Response (3)**
>
> **A2 (Continued):**
> - \(c) Our data composition ratio and experiments refute the possibility that the disconnect is due to DriveMind's distributional bias. We employed square-root sampling to ensure a balanced representation of driving behaviors (Eq. (1)). This produced a balanced distribution of difficulty levels (33.9% Hard, 42.0% General, 24.1% Simple). We have emphasized this on line 208 of page 4 in the revised version and have given the detailed scenario distribution in Appendix A3.
>
> Furthermore, we have categorized the 14 test scenario types into the same three difficulty levels:
> - Hard: High-dynamic scenarios (e.g., high lateral acceleration).
> - General: Standard rule-following scenarios (e.g., following lane with lead).
> - Simple: Low-dynamic scenarios (e.g., stationary in traffic).
>
> We evaluated the standard agent (CoT), the blind agent (Plan_NoV), and the prior-deprived agent (CoT_NoPri) across these categories. The detailed results are presented in the table below:
>
>
> | Category | Scenario | Open-Loop Score CoT | Open-Loop Score CoT_NoPri | Open-Loop Score Plan_NoV | Close-Loop Score CoT | Close-Loop Score CoT_NoPri | Close-Loop Score Plan_NoV |
> | :--- | :--- | ---: | ---: | ---: | ---: | ---: | ---: |
> | **Hard** | high_lateral_acceleration | 89.69 | 40.74 | 90.04 | 90.31 | 66.18 | 90.51 |
> | | behind_long_vehicle | 98.31 | 67.25 | 98.57 | 99.28 | 77.74 | 99.45 |
> | | changing_lane | 92.33 | 27.48 | 92.40 | 85.96 | 67.45 | 86.08 |
> | | starting_left_turn | 86.98 | 36.94 | 87.26 | 82.93 | 69.91 | 83.41 |
> | | starting_right_turn | 88.64 | 42.56 | 88.94 | 84.63 | 68.14 | 85.49 |
> | | near_multiple_vehicles | 93.42 | 42.72 | 93.51 | 92.79 | 81.82 | 92.67 |
> | | traversing_pickup_dropoff | 89.80 | 51.35 | 89.99 | 84.03 | 69.27 | 84.26 |
> | **Hard Avg** | | **91.31** | **44.15** | **91.53** | **88.56** | **71.50** | **88.84** |
> | **$\Delta$ (Diff)** | *(CoT - CoT_NoPri)* | **47.16** | | | **17.06** | | |
> | **General** | waiting_for_pedestrian_to_cross | 90.14 | 44.68 | 90.81 | 95.27 | 82.76 | 95.06 |
> | | starting_straight_traffic_light | 91.84 | 42.78 | 92.09 | 93.57 | 81.49 | 93.26 |
> | | stopping_with_lead | 93.03 | 67.84 | 94.87 | 94.14 | 81.75 | 94.35 |
> | | following_lane_with_lead | 95.16 | 67.07 | 96.14 | 99.98 | 91.49 | 99.98 |
> | **General Avg** | | **92.54** | **55.59** | **93.48** | **95.74** | **84.37** | **95.66** |
> | **$\Delta$ (Diff)** | *(CoT - CoT_NoPri)* | **36.95** | | | **11.37** | | |
> | **Simple** | low_magnitude_speed | 92.71 | 68.73 | 92.61 | 95.26 | 85.24 | 95.37 |
> | | stationary_in_traffic | 99.29 | 94.47 | 99.08 | 99.06 | 97.74 | 98.93 |
> | | high_magnitude_speed | 94.55 | 67.86 | 94.56 | 96.13 | 87.72 | 96.78 |
> | **Simple Avg** | | **95.52** | **77.02** | **95.42** | **96.82** | **90.23** | **97.03** |
> | **$\Delta$ (Diff)** | *(CoT - CoT_NoPri)* | **18.50** | | | **6.58** | | |
>
> The "blind" Plan_NoV agent achieves parity with the multimodal CoT agent even in Hard scenarios (91.53 vs. 91.31). This effectively refutes the hypothesis that "Plan NoV" succeeds only because the tasks are simple; rather, it demonstrates that textual priors provide a sufficiently robust shortcut to solve even complex intersection tasks without visual input.
>
> The performance gap between the standard agent and the prior-removed agent ($\Delta$) widens significantly as complexity increases (dropping by 47.16 points in Hard scenarios vs. 18.50 points in Simple ones). This suggests that as driving tasks become more challenging, the agent does not rely more on vision, but instead leans more heavily on the shortcut provided by textual priors.
>
> These stratified results confirm that the observed disconnect is a fundamental property of the model's learning strategy, not an artifact of the dataset. We have included this detailed breakdown in Page 20, Appendix A.10 of the revised paper to strengthen our findings.

---

> ### Author Response · Authors · 2025-11-21
> **Author Response (4)**
>
> > W3 \& Q1: Limited experimental scope and generalizability: The primary experiments focus on 7B-parameter models (Qwen2.5-VL, LLaVA-1.6) and an OmniDrive reimplementation. This raises several concerns: (a) Model scale: The findings may not generalize to larger VLMs (like 32B+), which may exhibit different reasoning capabilities and attention patterns. Testing at least one larger model would strengthen generalizability claims; (b) Proprietary models: The absence of closed-source model evaluation (e.g., GPT-4o as a driving agent) makes it unsure whether this is an artifact of open-source VLM training or a broader issue.
> >
> > Is the reasoning-planning disconnect specific to this model scale and Drivemind dataset, or does it generalize to larger models and other training recipes? Can the authors test larger open-source models or evaluate closed-source models in zero-shot settings to demonstrate broader applicability?
>
> **A3:** The concerns regarding model scale, dataset generation, and training recipes have been thoroughly addressed through the additional supplementary experiments:
>
>  We conduct zero-shot experiments on the significantly larger GPT-4o. Under the Causal Probe's perturbation, GPT-4o correctly articulate the necessary reasoning (e.g., "The ego-vehicle is in a lane moving straight"), yet the final predicted trajectory still succumb to the falsified textual prior, demonstrating that the reasoning-planning disconnect generalizes beyond the 7B parameter scale. Quantitatively, testing Causal Probe with $\delta=0.1$ on GPT-4o reveals substantial failure:
>
>
> | **Model** | **MFD ↓** | **RID ↓** |
> |--------------|-----------:|---------------------------:|
> | **GPT-4o**  | 2.55 | 1.35 |
>
> From the results, the MFD reached 2.25m, with a RID of 1.35. This large displacement confirms the reliance on textual priors.  The zero-shot evaluation on GPT-4o serves as the necessary closed-source model assessment, confirming that the disconnect is a fundamental issue across both open-source and proprietary, large-scale VLM architectures. We have added the experiments on GPT-4o in the newly introduced Section 4.5 on page 10 to demonstrate that our conclusions can be extended to larger models. We are also training a larger 32B-parameter driving agent (Qwen2.5-vl-32b, estimated 30 days on 8x A100-40G GPUs). These results will be included in the final manuscript.
>
> DriveMind leverages nuPlan, providing significantly richer scene semantics and interaction types than preceding benchmarks such as OmniDrive and DriveCoT (the table in A2). Crucially, our dataset ensures high fidelity: all core perception components (lane topology, object detection) are derived from nuPlan's high-fidelity ground truth rather than estimated computation. Furthermore, supervision for the planning task is provided by dense 5Hz ground-truth trajectory points, which are vital for precise, high-frequency planning tasks.
>
> The modular design of DriveMind enables seamless generalization across training protocols, demonstrating that our findings are not confined to a single format. Our standard setup (Input $\rightarrow$ CoT + Planning) directly mirrors the training recipe used by DriveCoT. By decomposing the modular CoT, the data structure aligns with the multi-task VQA approach used by OmniDrive. We have further conducted supplementary experiments using a multi-turn training composition (Input $\rightarrow$ CoT; Input + CoT $\rightarrow$ Planning). We note the trained agents as CoT\_Seq and CoT\_Seq\_NoPri, and the 3s planning scores of them are shown in the table below:
>
> | Agent | Open-Loop Score ↑| Close-Loop Score ↑|
> | :--- | :---: | :---: |
> | CoT_Seq | 91.64 | 91.45 |
> | CoT_Seq_NoPri | 49.61 | 74.62 |
>
> In this setting, the persistent disconnect still remains, confirming that the failure mechanism is robust across different training paradigms.
>
> Furthermore, we find the issue to be universal and robust to different optimization strategies. We have evaluated three distinct paradigms: (1) Supervised Fine-Tuning (SFT), which exhibits the baseline failure mode; (2) Counterfactual Reasoning (Omnidrive), where even the state-of-the-art Omnidrive agent, despite explicitly incorporating counterfactual reasoning to reduce prior dependence, collapses when textual priors are removed (Omnidrive_NoPri), revealing a persistent reliance on the shortcut; and (3) Reinforcement Learning (GRPO), where experiments using Group Relative Policy Optimization, a method designed to strengthen the reasoning–planning linkage, fail to mitigate the shortcut behavior. Together, these results demonstrate that the reasoning–planning disconnect persists across advanced policy optimization methods and is inherent to current VLM training recipes.

---

> ### Author Response · Authors · 2025-11-21
> **Author Response (5)**
>
> > Q2: Given that textual priors enable shortcut learning, did the authors experiment with alternative input designs or tasks that reduce prior specificity—such as coarse-grained navigation, noisy ego states, or prompts that require explicit visual grounding? This would clarify whether the issue is fundamental or can be mitigated through careful task design.
>
> **A4:** We appreciate this highly relevant line of inquiry, as experimenting with alternative input designs directly tests the nature of the model's reliance on textual priors.
>
> First, we clarify that our current training paradigm already requires explicit visual grounding. The core objective in our prompt is to require explicit visual grounding (i.e., object detection, lane detection). Our persistent findings confirm that such explicit instructions are insufficient to solve the disconnect; the shortcut remains the primary action driver.
>
> Second, we conduct a supplementary experiment using an alternative input design, employing noisy ego states. We train an agent, CoT\_noise, by introducing random noise ($\in [-0.2, 0.2]$) to the ego states (location and velocity) during training.
>
> | Agent | Open-Loop Score ↑| Close-Loop Score ↑|
> | :--- | :---: | :---: |
> | CoT | 92.56 | 92.38 |
> | CoT_Noise | 87.27 | 87.84 |
>
> We find that the overall planning ability of CoT\_noise decrease significantly compared to the original CoT agent. For instance, the Closed-Loop Score dropped from  92.38 to 87.84. This performance degradation strongly suggests that the model heavily relies on the precise, noise-free prior for learning, reinforcing the presence of the textual shortcut.
>
> While noise augmentation offered some degree of resistance against prior perturbations, CoT\_noise is still highly sensitive to the causal probe.
>
> | Agent | MFD ↓| RID ↓|
> | :--- | :---: | :---: |
> | CoT | 3.751 | 2.04 |
> | CoT_Noise | 0.982 | 0.61 |
>
> Testing at $\delta=0.08$ showed an average planning error (MFD) of approximately $0.98\text{m}$. This confirms that the fundamental dependency on the prior is not solved by simple noise injection. In summary, simple solutions like explicit visual grounding prompts or input noise augmentation do not solve the fundamental reasoning-planning disconnect. The failure of these standard mitigating measures, coupled with the universality of the decoupling across SOTA training recipes (Omnidrive, GRPO), underscores that the issue is a fundamental vulnerability of the VLM training paradigm.
>
> The reviewer's suggestion aligns perfectly with our planned future work (Section 5), specifically regarding "Breaking Shortcuts via Contrastive Learning." We appreciate the insight, as it provides further valuable direction for our future work.
>
> > Q3: The causal probe perturbs textual priors. Have the authors tried perturbing visual inputs (e.g., masking, noise, scene replacement)? If planning truly relies on textual shortcuts, visual perturbations should have minimal impact.
>
> **A5:** Testing visual perturbations is the inverse check for the reliance on textual shortcuts. If planning truly relies on priors, visual degradation should have minimal impact on the planning outcome.
>
> Our initial results have already validated this hypothesis comprehensively: the fact that the Plan_NoV agent (complete removal of visual input) performs comparably to the CoT agent (full visual input) establishes that the planning output is fundamentally insensitive to the complete absence of visual information. This is the most extreme form of visual perturbation possible.
>
> To further solidify this finding, we conduct two additional visual perturbation experiments:
> | Agent | Open-Loop Score ↑| Close-Loop Score ↑|
> | :--- | :---: | :---: |
> | CoT | 92.56 | 92.38 |
> | CoT_V_Crop_Noise | 91.86 | 90.79 |
> | CoT_V_Replace | 92.36 | 92.27 |
> - Scene Replacement (CoT_V_Replace): Randomly replacing the correct visual scene with a contradictory one result in minimal performance change (e.g., Close-Loop score dropped only from 92.38 to 92.27). This confirms that the model ignores the content of the image when making planning decisions.
> - Severe Noise (CoT_V_Crop_Noise): Even under severe cropping and heavy Gaussian noise, the model maintains high closed-loop compliance (90.79). The slightly larger drop compared to CoT_V_Replac is attributed primarily to non-semantic artifacts introduced by the severe format mismatch (incorrect input image size relative to the training data), which further proves the model is not relying on the image's semantic content.
>
> These supplementary results comprehensively confirm that the planning ability is effectively visually inert; it functions by relying solely on the textual priors, making fine-grained visual masking or noise perturbations redundant for confirming the existence of the shortcut pathway. We have added these experimental results on visual input perturbation in Appendix A.11 on page 20 to strengthen the persuasiveness of our paper.

---

> ### Author Response · Authors · 2025-11-21
> **Author Response (6)**
>
> In summary, we sincerely appreciate the reviewer's time and careful evaluation of our paper, and we are grateful for the constructive comments that have undoubtedly contributed to improving our work, such as inspiring us to make the causal probe into a reusable benchmark. Should the reviewer have any further questions, we would be happy to discuss them at any time.

---

### Official Review · Reviewer_Q5kf · 2025-11-01

**Soundness:** 3
**Presentation:** 2
**Contribution:** 2
**Rating:** 4
**Confidence:** 4

**Summary:**

This paper explores the causal relationship between reasoning and planning in VLM based autonomous driving agents. To facilitate rigorous analysis, the authors introduce the DriveMind dataset, which is specifically designed for causal investigation. The work applies a variety of diagnostic methods—including systematic ablations, perturbation-based causal probes, and sequence-level attention analysis—to comprehensively study how current VLMs achieve driving performance. Across multiple experiments and baselines, the authors find strong evidence for a 'reasoning-planning disconnect,' where agents predominantly rely on textual priors for planning and CoT reasoning is often non-causal.

**Strengths:**

1. The paper identifies and empirically validates a central issue in the development of autonomous driving agents: the disconnect between reasoning and planning in VLM-based models.

2. The paper introduces DriveMind, a large-scale nuPlan-based dataset with plan-aligned CoT and modular inputs, enabling rigorous causal analysis and systematic ablations for vision-language driving models.

3. The proposed Causal Probe offers a novel, training-free diagnostic method to identify shortcut reliance, providing a practical tool for evaluating the causal robustness of driving agents.

**Weaknesses:**

1. COT seems to contain many dimensions of scene information (Traffic Light Analysis, Weather Analysis, etc.), but it is unclear whether every case analyzes the same set of dimensions, or whether the CoT content is truly focused on those features most strongly related to the subsequent driving action. The potential redundancy in CoT construction might reduce the relevance and specificity of training signals.

2. In Section 3.1, it is mentioned that “GPT-4.1 is tasked with explaining the causal logic behind the expert trajectory,” but Figure 5 in the appendix shows a prompt where the model is required to generate both the CoT and the trajectory sequence simultaneously. This could lead to confusion about the source of trajectory ground truth, and clarification from the authors would be appreciated.

3. In Section 4.2, the text notes that “the Plan NoV agent, effectively ‘driving blind’ without any visual input, performs on par with the CoT agent.” However, such results could arise from ordinary driving scenarios and basic behaviors. The paper does not provide a detailed distribution of driving scenarios or behaviors within the DriveMind dataset, nor a case study for particularly challenging situations. Supplementary statistics or case studies on scenario diversity and difficulty would help address concerns about evaluation rigor.

4. In Section 4.4, the experiments on the causal probe present only a small set of illustrative cases, which limits the persuasiveness of the findings. It is recommended to include quantitative metrics and analysis to demonstrate that the conclusions hold across a broad range of scenarios.

**Questions:**

1. COT seems to contain many dimensions of scene information (Traffic Light Analysis, Weather Analysis, etc.), but it is unclear whether every case analyzes the same set of dimensions, or whether the CoT content is truly focused on those features most strongly related to the subsequent driving action. The potential redundancy in CoT construction might reduce the relevance and specificity of training signals.

2. In Section 3.1, it is mentioned that “GPT-4.1 is tasked with explaining the causal logic behind the expert trajectory,” but Figure 5 in the appendix shows a prompt where the model is required to generate both the CoT and the trajectory sequence simultaneously. This could lead to confusion about the source of trajectory ground truth, and clarification from the authors would be appreciated.

3. In Section 4.2, the text notes that “the Plan NoV agent, effectively ‘driving blind’ without any visual input, performs on par with the CoT agent.” However, such results could arise from ordinary driving scenarios and basic behaviors. The paper does not provide a detailed distribution of driving scenarios or behaviors within the DriveMind dataset, nor a case study for particularly challenging situations. Supplementary statistics or case studies on scenario diversity and difficulty would help address concerns about evaluation rigor.

4. In Section 4.4, the experiments on the causal probe present only a small set of illustrative cases, which limits the persuasiveness of the findings. It is recommended to include quantitative metrics and analysis to demonstrate that the conclusions hold across a broad range of scenarios.

---

> ### Author Response · Authors · 2025-11-21
> **Author Response (1)**
>
> We sincerely appreciate the reviewer for his/her valuable time and constructive feedback. We would like to address all of the points raised in this careful review and provide additional context or clarification that we believe could contribute to the reviewer's acceptance of our paper. We have also uploaded our revised pdf paper and marked important revisions as blue. Please find our detailed responses below in conjunction with the revised paper.:
>
> > W1 \& Q1: COT seems to contain many dimensions of scene information (Traffic Light Analysis, Weather Analysis, etc.), but it is unclear whether every case analyzes the same set of dimensions, or whether the CoT content is truly focused on those features most strongly related to the subsequent driving action. The potential redundancy in CoT construction might reduce the relevance and specificity of training signals.
>
> **A1:** We respectfully clarify that the structure of our CoT is intentionally designed to balance a "comprehensive safety check" with "focused decision-making," and we have empirically verified that redundancy is not the cause of the observed disconnect.
>
>  While the dimensions of the CoT (e.g., Traffic Light Analysis, Weather) are fixed to function as a safety checklist, preventing the model from overlooking critical categories, the content within these sections is strictly dynamic and scenario-dependent. For instance, as shown in our dataset examples, if no relevant signal is present, the analysis explicitly states: "No traffic lights detected in the current scenario". This ensures that the CoT reflects the actual scene status rather than generic boilerplate.
>
> As detailed in Appendix A.2 , the CoT concludes with a "Macro Driving Decision" section. This module is specifically designed to filter the preceding observations and synthesize only the features most strongly related to the driving action (e.g., explicitly citing a "gentle leftward curve" as the rationale for steering ).
>
> To definitively rule out the hypothesis that CoT redundancy dilutes the training signal, we conduct an additional experiment. We train a new set of agents, CoT_D and CoT_D_NoPri, where the CoT is stripped of all "environment checks" and retain only the  Object Detection analysis (highly relevant to planning). The performance patterns of CoT_D and CoT_D_NoPri are nearly identical to the original CoT and CoT_NoPri reported in the paper. The simplified CoT does not improve the reasoning-planning link; the model still collapses when textual priors are removed.
>
> This provides robust evidence that the disconnect is not a byproduct of CoT redundancy or lack of specificity.
>
>
> > W2 \& Q2: In Section 3.1, it is mentioned that “GPT-4.1 is tasked with explaining the causal logic behind the expert trajectory,” but Figure 5 in the appendix shows a prompt where the model is required to generate both the CoT and the trajectory sequence simultaneously. This could lead to confusion about the source of trajectory ground truth, and clarification from the authors would be appreciated.
>
> **A2:** We sincerely apologize for the confusion arising from the presentation of our figures and the description of the data generation process. We wish to provide a definitive clarification regarding the source of the ground truth trajectories.
>
> We explicitly confirm that the trajectory ground truth in the DriveMind dataset is derived entirely from the nuPlan expert driving logs. These trajectories are not generated by GPT-4.1. GPT-4.1 is employed solely to generate the perceptual analysis (e.g., weather) and to articulate the causal logic explaining the expert's decision. The previous Figure 5 displays a final training sample from our dataset. This is an example of data for Supervised-Finetuning a driving agent, not for GPT-4.1 to generate CoT. We clarify that the values in the \<answer\> block are the ground-truth trajectories extracted directly from nuPlan, while the content within the \<think\> block represents the plan-aligned CoT generated by GPT-4.1.
> The previous Figure 6  illustrates the prompt used during the data generation phase to elicit the CoT from GPT-4.1.
>
> To prevent further misunderstanding, we have explicitly stated in the revised main text that all target trajectories originate strictly from the nuPlan dataset and are not model-generated (Page 4, Line 198). We have also swaped the order of the previous Figures 5 and 6, presenting the generation prompt (currently Fig. 5) before the resulting dataset sample (currently Fig. 6). We believe this will provide a more logical flow and clarify the data construction pipeline for readers.

---

> ### Author Response · Authors · 2025-11-21
> **Author Response (2)**
>
> > W3 \& Q3: In Section 4.2, the text notes that “the Plan NoV agent, effectively ‘driving blind’ without any visual input, performs on par with the CoT agent.” However, such results could arise from ordinary driving scenarios and basic behaviors. The paper does not provide a detailed distribution of driving scenarios or behaviors within the DriveMind dataset, nor a case study for particularly challenging situations. Supplementary statistics or case studies on scenario diversity and difficulty would help address concerns about evaluation rigor.
>
> **A3:** To rigorously answer this question, we conducted a comprehensive stratified evaluation and analyzed the scenario distribution within the DriveMind dataset.
>
> We confirm that our dataset is not dominated by trivial cases. We employ square-root sampling to ensure a balanced representation of driving behaviors (Eq. (1)). This produced a balanced distribution of difficulty levels (33.9% Hard, 42.0% General, 24.1% Simple). We have emphasized this on line 208 of page 4 in the revised version and have given the detailed scenario distribution in Appendix A3.
>
> Furthermore, we have also categorized the 14 scenario types in the test set into the same three difficulty levels:
>
> - Hard: High-dynamic scenarios (e.g., high lateral acceleration, near multiple vehicles).
> - General: Standard rule-following scenarios (e.g., following lane with lead).
> - Simple: Low-dynamic scenarios (e.g., stationary in traffic).
>
> We evaluated the standard agent (CoT), the blind agent (Plan_NoV), and the prior-deprived agent (CoT_NoPri) across these categories. The detailed results are presented in the table below:
> | Category | Scenario | Open-Loop Score CoT | Open-Loop Score CoT_NoPri | Open-Loop Score Plan_NoV | Close-Loop Score CoT | Close-Loop Score CoT_NoPri | Close-Loop Score Plan_NoV |
> | :--- | :--- | ---: | ---: | ---: | ---: | ---: | ---: |
> | **Hard** | high_lateral_acceleration | 89.69 | 40.74 | 90.04 | 90.31 | 66.18 | 90.51 |
> | | behind_long_vehicle | 98.31 | 67.25 | 98.57 | 99.28 | 77.74 | 99.45 |
> | | changing_lane | 92.33 | 27.48 | 92.40 | 85.96 | 67.45 | 86.08 |
> | | starting_left_turn | 86.98 | 36.94 | 87.26 | 82.93 | 69.91 | 83.41 |
> | | starting_right_turn | 88.64 | 42.56 | 88.94 | 84.63 | 68.14 | 85.49 |
> | | near_multiple_vehicles | 93.42 | 42.72 | 93.51 | 92.79 | 81.82 | 92.67 |
> | | traversing_pickup_dropoff | 89.80 | 51.35 | 89.99 | 84.03 | 69.27 | 84.26 |
> | **Hard Avg** | | **91.31** | **44.15** | **91.53** | **88.56** | **71.50** | **88.84** |
> | **$\Delta$ (Diff)** | *(CoT - CoT_NoPri)* | **47.16** | | | **17.06** | | |
> | **General** | waiting_for_pedestrian_to_cross | 90.14 | 44.68 | 90.81 | 95.27 | 82.76 | 95.06 |
> | | starting_straight_traffic_light | 91.84 | 42.78 | 92.09 | 93.57 | 81.49 | 93.26 |
> | | stopping_with_lead | 93.03 | 67.84 | 94.87 | 94.14 | 81.75 | 94.35 |
> | | following_lane_with_lead | 95.16 | 67.07 | 96.14 | 99.98 | 91.49 | 99.98 |
> | **General Avg** | | **92.54** | **55.59** | **93.48** | **95.74** | **84.37** | **95.66** |
> | **$\Delta$ (Diff)** | *(CoT - CoT_NoPri)* | **36.95** | | | **11.37** | | |
> | **Simple** | low_magnitude_speed | 92.71 | 68.73 | 92.61 | 95.26 | 85.24 | 95.37 |
> | | stationary_in_traffic | 99.29 | 94.47 | 99.08 | 99.06 | 97.74 | 98.93 |
> | | high_magnitude_speed | 94.55 | 67.86 | 94.56 | 96.13 | 87.72 | 96.78 |
> | **Simple Avg** | | **95.52** | **77.02** | **95.42** | **96.82** | **90.23** | **97.03** |
> | **$\Delta$ (Diff)** | *(CoT - CoT_NoPri)* | **18.50** | | | **6.58** | | |
>
> The "blind" Plan_NoV agent achieves parity with the multimodal CoT agent even in Hard scenarios (91.53 vs. 91.31). This effectively refutes the hypothesis that "Plan NoV" succeeds only because the tasks are simple; rather, it demonstrates that textual priors provide a sufficiently robust shortcut to solve even complex intersection tasks without visual input.
>
> The performance gap between the standard agent and the prior-removed agent ($\Delta$) widens significantly as complexity increases (dropping by 47.16 points in Hard scenarios vs. 18.50 points in Simple ones). This suggests that as driving tasks become more challenging, the agent does not rely more on vision, but instead leans more heavily on the shortcut provided by textual priors.
>
> These stratified results confirm that the observed disconnect is a fundamental property of the model's learning strategy, not an artifact of dataset simplicity. We have included this detailed breakdown in Page 20, Appendix A.10 of the revised paper to strengthen the evaluation rigor.

---

> ### Author Response · Authors · 2025-11-21
> **Author Response (3)**
>
> > W4 \& Q4: In Section 4.4, the experiments on the causal probe present only a small set of illustrative cases, which limits the persuasiveness of the findings. It is recommended to include quantitative metrics and analysis to demonstrate that the conclusions hold across a broad range of scenarios.
>
> **A4:** We acknowledge the concern that illustrative cases alone may limit the persuasiveness of our findings. To demonstrate the generalizability of the reasoning-planning disconnect, we performed a large-scale **Quantitative Sensitivity Analysis** using our causal probe.
>
> To better quantify the model's sensitivity, we report the Mean Final Deviation (MFD) (m) of the trajectory endpoint and the Relative Intervention Degree (RID). The MFD is defined as:$$MFD = \frac{\sum_{i=1}^{n} |Original\\\_Final\\\_xy_{i} - Perturbed\\\_Final\\\_xy_{i}| }{n},$$ where n is the number of the test samples.
>
> RID is defined as the ratio of the difference between a sample's output before and after perturbation to the theoretical error that would occur if the model perfectly followed the perturbation (i.e., $\delta \cdot v_i \cdot T$):$$RID =\frac{1}{n}\sum_{i=1}^{n}\frac{ |Original\\\_Final\\\_xy_{i} - Perturbed\\\_Final\\\_xy_{i}|}{\delta \cdot v_i \cdot T},$$ where $v_{i}$ is the ego-velocity of the i-th sample and $T$ is the prediction horizon. This normalized comparison enables RID to quantitatively measure the extent to which a model relies on textual priors: RID=1 indicates complete reliance on the error in textual priors, RID>1 indicates amplification of the error, and RID<1 indicates the ability to correct the error through visual reasoning. This provides a scale-independent, robust metric that allows for cross-scenario comparisons.
>
> We expand our analysis by conducting a sensitivity analysis on the CoT_grpo agent. We utilize a test set of 300 randomly sampled scenarios and vary the perturbation magnitude $\delta$ from 0.02 to 0.3:
>
> | $\delta$       | 0.02  | 0.06  | 0.08  | 0.10  | 0.30   |
> |----------------|-------|-------|-------|-------|--------|
> | MFD  ↓          | 0.55 | 2.17 | 3.75 | 5.89 | 15.46 |
> | RID ↓  | 1.00  | 1.47  | 2.04  | 2.17  | 2.04   |
>
>
> The quantitative results reveal a distinct two-phase failure mode:
> - Immediate Fragility: Even at a minimal perturbation of $\delta=0.06$, the MFD jumps to 2.17m, indicating that the model lacks the visual robustness to correct for subtle textual deviations.
> - Saturation without Correction: As $\delta$ increases, the MFD grows monotonically (reaching 15.46m at $\delta=0.3$). The RID stabilizes around ~2.1, which implies that the model's "wrong output" has saturated. It consistently generates erroneous trajectories driven by the falsified prior rather than reverting to the correct visual ground truth.
>
> These results quantitatively prove that the "disconnect" and "prior reliance" are systematic behaviors across the dataset, not isolated anecdotes. We have included our quantitative metrics on line 332 of page 7 in the revised version and updated the results of this probe experiment on line 465 of page 9.
>
> Overall, we thank the reviewer for helping us strengthen our work by pointing out several areas for clarification and improvement. The reviewers' comments have helped make our paper clearer and more coherent, and inspire us to design more rigorous stratified experiments to validate our findings, as well as to develop quantifiable, scalable causal probes. Should the reviewer have any further questions or require additional discussion, we welcome the opportunity to clarify our findings.

---

### Official Review · Reviewer_zZua · 2025-11-03

**Soundness:** 3
**Presentation:** 3
**Contribution:** 3
**Rating:** 6
**Confidence:** 4

**Summary:**

This paper investigates whether the reasoning produced by Vision-Language driving agents truly causes their planning behavior. The authors build DriveMind, a nuPlan-based dataset with plan-aligned Chain-of-Thought reasoning, and perform systematic ablation and perturbation studies on several VLM backbones. The main finding is striking: removing reasoning or vision barely affects planning, but removing textual priors causes a sharp drop—suggesting that current models rely heavily on shortcuts rather than genuine reasoning.

**Strengths:**

1. First large-scale causal analysis explicitly targeting the reasoning-planning link in VLM-driving agents.

2. Multiple baselines (SFT, GRPO, OmniDrive, Llava) and clear ablation logic (with/without CoT, priors, vision).

3. Sequence-level attention analysis and causal probe provide intuitive visual evidence of shortcut learning.

4. Findings question a core assumption in explainable autonomous-driving research and open a new evaluation perspective.

**Weaknesses:**

1. The “disconnect” remains correlational; causal claims rely on ablation heuristics rather than formal interventions or counterfactual reasoning frameworks.

2. All tests are in simulation (nuPlan) with no real-world validation or human-driving baselines.

3. Other explanations (data bias, network capacity, token-level co-training) are not empirically ruled out.

**Questions:**

1. How consistent are these findings across different scene complexities (e.g., intersections vs highways)? Stratify evaluation by scenario complexity to verify whether shortcut reliance strengthens under challenging conditions will be a good bonus.

2. Could token-level leakage between reasoning and planning outputs (e.g., implicit trajectory hints in CoT) explain part of the observed independence?

3. How sensitive are the causal-probe results to the perturbation magnitude δ? It's better to see some experiments between the results and the perturbation level.

---

> ### Author Response · Authors · 2025-11-21
> **Author Response (1)**
>
> We sincerely appreciate the reviewer's thorough review of our paper and the positive feedback. We have carefully addressed all concerns and incorporated the relevant clarifications and revisions into the updated version of the paper. The detailed responses are provided below, and the corresponding changes can be found in the revised manuscript.
>
> >W1: The “disconnect” remains correlational; causal claims rely on ablation heuristics rather than formal interventions or counterfactual reasoning frameworks.
>
> **A1:** We respectfully posit that our methodology establishes causal relationships through rigorous structural interventions rather than mere correlation.
> - First, our ablation studies (currently Table 3) function as formal interventions on the causal graph nodes. By systematically severing the visual input ($V$) and observing that the output ($Planning$) remains statistically invariant, we demonstrate that $V$ is not acting as a causal mediator for planning in current architectures.
> - Second, our causal probe diagnostic framework uses counterfactual reasoning. For example, suppose the image clearly shows a left-curving road. In the counterfactual test, we keep the image fixed but alter the textual history to falsely suggest the vehicle has been merging from the right. If the agent truly relied on the visual scene, it should still plan a left turn. However, our results show the opposite: even when the reasoning of the visual input dictates a left turn, the agent follows the falsified textual history and produces a rightward trajectory. This demonstrates that the planning output is directly caused by the prior ($Pri$), not the visual scene ($V$), when the two conflict.
>
> > W2: All tests are in simulation (nuPlan) with no real-world validation or human-driving baselines.
>
> **A2:** We respectfully clarify that our setup is grounded in high-fidelity real-world data and aligns with established standards in the field.
>
> - First, unlike simulation platforms such as CARLA, nuPlan is built entirely from real sensor logs and trajectories collected by Motional’s autonomous fleet. The "simulation" in our context only refers to the replay of authentic human driving data in the evaluation, ensuring that both the visual inputs and traffic dynamics reflect the complexity of the real world. Furthermore, the ground-truth trajectories used as targets in our dataset represent expert human driving, which inherently serves as the human baseline for our analysis.
> - Second, our evaluation protocol is consistent with recent prominent studies in VLM driving agents, such as Omnidrive and DriveLM. We benchmark against these representative methods to ensure our findings are positioned relevantly within the current literature. Importantly, our main contribution is diagnosing an internal failure mode. Because this issue stems from how the model processes information (e.g., shortcut learning from priors), it can be rigorously isolated and verified using high-fidelity logs.
> - Third, our experiments reveal that current VLM agents can achieve high performance scores even when visual inputs are completely removed (the "Plan\_NoV" agent). Given this proven "blindness" and reliance on shortcuts, deploying such models onto a physical vehicle for real-world validation would be unsafe and ethically precarious.
>
> We hope this clarifies the validity of our evaluation platform and the rationale behind our design. We are happy to provide further details if needed.

---

> ### Author Response · Authors · 2025-11-21
> **Author Response (2)**
>
> > W3: Other explanations (data bias, network capacity, token-level co-training) are not empirically ruled out.
>
> **A3:** We conducted additional experiments to rule out token-level leakage, data bias, and limited model capacity.
> - Data bias: To ensure our findings are not caused by dataset imbalance, we carefully constructed the DriveMind dataset using square-root weighted stratified sampling over 61 scenarios (Eq. (1)). This produced a balanced distribution of difficulty levels (33.9% Hard, 42.0% General, 24.1% Simple). We have emphasized this on line 208 of page 4 in the revised version and have given the detailed scenario distribution in Appendix A3. DriveMind also uses nuPlan’s rich semantic annotations (e.g., 3D objects, traffic lights) to generate high-quality, plan-aligned CoT. Since the reasoning–planning disconnect persists even under this balanced and semantically rich setup, the issue clearly stems from the model’s learning behavior (i.e., shortcutting), not from dataset bias. (For more dataset details, please refer to our answers to  Q1.)
> - Network Capacity: The disconnect is not due to limited capacity. We furhter test our casual probe on GPT-4o in a zero-shot setting. Although GPT-4o correctly reasons that the vehicle should drive straight, its predicted trajectory still drifted in response to the perturbation. Our quantitative results for the causal probe with $\delta$=0.1 on GPT-4o are shown in the table below (refer to our A3 for detailed definitions of MFD and RID metrics).
>
>     | **Model** | **MFD ↓** | **RID ↓** |
>     |--------------|-----------:|---------------------------:|
>     | **GPT-4o**  | 2.55 | 1.35 |
>
>     This shows the issue generalizes even to top-tier large VLM models. We have added the experiments on GPT-4o in the newly introduced Section 4.5 on page 10 to demonstrate that our conclusions can be extended to larger models. To further confirm, we are training a larger 32B-parameter driving agent (Qwen2.5-vl-32b, estimated 30 days on 8x A100-40G GPUs). The related results will be included in the final manuscript.
>
>
> - Token-level Co-training: We also have verified that the model is not "cheating" by extracting trajectory hints from CoT tokens. Table 3 shows that the "Plan" agent, which is trained without any reasoning tokens, performs almost identically to the "CoT" agent (Closed-Loop Scores: 97.38 vs. 97.46 at 1s). Since the "Plan" agent never sees reasoning text, its performance cannot come from token-level leakage. This indicates that both agents rely on textual priors in similar ways, and the CoT does not actually guide the planning module.

---

> ### Author Response · Authors · 2025-11-21
> **Author Response (3)**
>
> >Q1: How consistent are these findings across different scene complexities (e.g., intersections vs highways)? Stratify evaluation by scenario complexity to verify whether shortcut reliance strengthens under challenging conditions will be a good bonus.
>
> **A4**: We stratify the 14 distinct evaluation types in the nuPlan Challenge test set into three complexity levels based on dynamic density and interaction difficulty.
>
> - Hard: High-dynamic scenarios involving complex interactions or edge cases (e.g., turning, lane changing).
> - General: Standard driving scenarios governed by clear traffic rules (e.g., following lane with lead, stopping at lights).
> - Simple: Low-dynamic or static scenarios (e.g., stationary in traffic, low magnitude speed).
>
> We measure the performance of the standard agent (CoT), the blind agent (Plan_NoV), and the prior-deprived agent (CoT_NoPri) across these categories. The results are summarized below:
>
> | Category | Scenario | Open-Loop Score CoT | Open-Loop Score CoT_NoPri | Open-Loop Score Plan_NoV | Close-Loop Score CoT | Close-Loop Score CoT_NoPri | Close-Loop Score Plan_NoV |
> | :--- | :--- | ---: | ---: | ---: | ---: | ---: | ---: |
> | **Hard** | high_lateral_acceleration | 89.69 | 40.74 | 90.04 | 90.31 | 66.18 | 90.51 |
> | | behind_long_vehicle | 98.31 | 67.25 | 98.57 | 99.28 | 77.74 | 99.45 |
> | | changing_lane | 92.33 | 27.48 | 92.40 | 85.96 | 67.45 | 86.08 |
> | | starting_left_turn | 86.98 | 36.94 | 87.26 | 82.93 | 69.91 | 83.41 |
> | | starting_right_turn | 88.64 | 42.56 | 88.94 | 84.63 | 68.14 | 85.49 |
> | | near_multiple_vehicles | 93.42 | 42.72 | 93.51 | 92.79 | 81.82 | 92.67 |
> | | traversing_pickup_dropoff | 89.80 | 51.35 | 89.99 | 84.03 | 69.27 | 84.26 |
> | **Hard Avg** | | **91.31** | **44.15** | **91.53** | **88.56** | **71.50** | **88.84** |
> | **$\Delta$ (Diff)** | *(CoT - CoT_NoPri)* | **47.16** | | | **17.06** | | |
> | **General** | waiting_for_pedestrian_to_cross | 90.14 | 44.68 | 90.81 | 95.27 | 82.76 | 95.06 |
> | | starting_straight_traffic_light | 91.84 | 42.78 | 92.09 | 93.57 | 81.49 | 93.26 |
> | | stopping_with_lead | 93.03 | 67.84 | 94.87 | 94.14 | 81.75 | 94.35 |
> | | following_lane_with_lead | 95.16 | 67.07 | 96.14 | 99.98 | 91.49 | 99.98 |
> | **General Avg** | | **92.54** | **55.59** | **93.48** | **95.74** | **84.37** | **95.66** |
> | **$\Delta$ (Diff)** | *(CoT - CoT_NoPri)* | **36.95** | | | **11.37** | | |
> | **Simple** | low_magnitude_speed | 92.71 | 68.73 | 92.61 | 95.26 | 85.24 | 95.37 |
> | | stationary_in_traffic | 99.29 | 94.47 | 99.08 | 99.06 | 97.74 | 98.93 |
> | | high_magnitude_speed | 94.55 | 67.86 | 94.56 | 96.13 | 87.72 | 96.78 |
> | **Simple Avg** | | **95.52** | **77.02** | **95.42** | **96.82** | **90.23** | **97.03** |
> | **$\Delta$ (Diff)** | *(CoT - CoT_NoPri)* | **18.50** | | | **6.58** | | |
>
> As observed in the table, the "blind" Plan_NoV agent achieves parity with (or slightly exceeds) the multimodal CoT agent across all complexity levels, including "Hard" scenarios. This confirms that the reasoning-planning disconnect is not limited to simple cases; even in complex turning scenarios, the model ignores visual cues in favor of priors. The performance gap between the standard agent and the prior-removed agent ($\Delta$ (CoT - CoT_NoPri)) serves as a proxy for "reliance on priors." Notably, this gap widens significantly as complexity increases: the Open-Loop score  of CoT_NoPri drops by 47.16 points in Hard scenarios compared to only 18.50 points in Simple scenarios. These results confirm that the model's reliance on shortcuts is not only consistent but effectively strengthens under challenging conditions. Rather than engaging in deeper visual reasoning for complex tasks, the agent leans more heavily on textual priors to solve the problem. We have added these stratified test results to Appendix A.10 on page 20 of the revised version.
>
> > Q2: Could token-level leakage between reasoning and planning outputs (e.g., implicit trajectory hints in CoT) explain part of the observed independence?
>
> **A5:** We appreciate this insightful question regarding the potential for information leakage. As detailed in our Response to W3 (specifically under "Token-level Co-training"), we have provided empirical evidence that effectively rules this out.

---

> ### Author Response · Authors · 2025-11-21
> **Author Response (4)**
>
> >Q3: How sensitive are the causal-probe results to the perturbation magnitude δ? It's better to see some experiments between the results and the perturbation level.
>
> **A6:** We expand our analysis by conducting a sensitivity analysis on the CoT_grpo agent. We utilize a test set of 300 randomly sampled scenarios and vary the perturbation magnitude $\delta$ from 0.02 to 0.3. To better quantify the model's sensitivity, we report the Mean Final Deviation (MFD) (m) of the trajectory endpoint and the Relative Intervention Degree (RID). The MFD is defined as:$$MFD = \frac{\sum_{i=1}^{n} |Original\\\_Final\\\_xy_{i} - Perturbed\\\_Final\\\_xy_{i}| }{n},$$ where n is the number of tested samples.
>
> RID is defined as the ratio of the difference between a sample's output before and after perturbation to the theoretical error that would occur if the model perfectly followed the perturbation (i.e., $\delta \cdot v_i \cdot T$):$$RID =\frac{1}{n}\sum_{i=1}^{n}\frac{ |Original\\\_Final\\\_xy_{i} - Perturbed\\\_Final\\\_xy_{i}|}{\delta \cdot v_i \cdot T},$$ where $v_{i}$ is the ego-velocity of the i-th sample and $T$ is the prediction horizon. This normalized comparison enables RID to quantitatively measure the extent to which a model relies on textual priors: RID=1 indicates complete reliance on the error in textual priors, RID>1 indicates amplification of the error, and RID<1 indicates the ability to correct the error through visual reasoning. This provides a scale-independent, robust metric that allows for cross-scenario comparisons.
>
> The perturbation test result on CoT_grpo is as below:
>
> | $\delta$       | 0.02  | 0.06  | 0.08  | 0.10  | 0.30   |
> |----------------|-------|-------|-------|-------|--------|
> | MFD  ↓          | 0.55 | 2.17 | 3.75 | 5.89 | 15.46 |
> | RID ↓  | 1.00  | 1.47  | 2.04  | 2.17  | 2.04   |
>
>
> The model exhibits extreme instability even at low perturbation levels. As $\delta$ increases from 0.02 to 0.06, the  MFD jumps significantly to 2.17m. This confirms that the agent is captured by the textual prior almost immediately, lacking the visual robustness to correct for even subtle semantic conflicts.  As $\delta$ increases further (0.08 to 0.3), the MFD continues to grow monotonically, reaching 15.46m at $\delta=0.3$. The RID initially rises but then plateaus or slightly decreases (e.g., it peaks at 2.17 for $\delta=0.1$ and drops to 2.04 for $\delta=0.3$). However, this plateau does not indicate increased robustness. Since the MFD continues to skyrocket and the RID is still larger than 1, the stabilization of the ratio implies that the model's "wrong output" has saturated. The agent is predicting extreme trajectories that exceed the boundaries of its learned manifold (the shortcuts it memorized), causing its reaction speed to lag behind the aggressive growth of the input perturbation.
>
> The results demonstrate that the reasoning-planning disconnect is pervasive. The model collapses at low perturbation magnitudes ($\delta=0.06$) and drifts into saturation at high magnitudes, proving that the textual prior acts as the dominant, uncorrected driver of planning behaviors. We have described our quantitative metrics on line 332 of page 7 in the revised version and updated the results of this probe experiment on line 465 of page 9.
>
>
>
> In summary, We sincerely thank reviewer zZua for the thorough review and constructive feedback. These comments guide us in better substantiating our conclusions (e.g., through stratified evaluation by scenario complexity) and inspire us to design a quantifiable, universal probe method. We extend our gratitude once again to this reviewer and welcome any potential discussion.

---

### Author Response · Authors · 2025-12-01
**Rebuttal Summary**

Dear PC, AC, and reviewers,

During the rebuttal period, we have addressed all reviewers’ concerns with detailed point-by-point responses and have revised the paper based on these constructive suggestions. We hereby provide a summary:

**Main refinements:**
>Refine causal probes into quantifiable benchmark metrics. (zZua Q3, Q5kf Q4/W4, Head W1)

We refine the lateral-offset causal probe by introducing two quantitative metrics: Mean Final Deviation (MFD), which measures the absolute endpoint deviation (m), and Relative Intervention Degree (RID), a normalized, scale-independent metric capturing a model’s sensitivity to textual priors. Together with our DriveMind dataset, these transform the causal probe into a reusable benchmark. We expand evaluation for perturbation amplitudes $\delta \in [0.02, 0.3]$ using MFD and RID, and the results under different $\delta$ consistently support our conclusions, validating the robustness of our findings. Definitions and updated results are provided in the revision (Page 7, Line 332; Page 9, Line 465 & Table 5).
>Add causal probe experiments on the large-scale commercial model GPT-4o. (zZua W3, Head W3/Q1)

We further test our causal probe on GPT-4o in a zero-shot setting, confirming that the reasoning-planning disconnect is not due to limited network capacity; it is a fundamental issue across both open-source and proprietary, large-scale VLM architectures. We have added the experiments on GPT-4o in the newly introduced Section 4.5 on page 10.
>Add detailed DriveMind data distributions and comparison with existing datasets. (zZua W3, Head W2 (a) (c))

We present the proportions of scenarios across difficulty levels in DriveMind (revised version line 208), with full per-scenario distributions provided in Appendix A.3. These statistics show that DriveMind is both balanced and semantically diverse, reinforcing that the observed reasoning–planning disconnect arises from model shortcutting, not dataset bias. Additionally, Table 1 now compares DriveMind with relevant datasets and highlights its unique combination of rich, structured semantic context derived from real-world logs and its modular design tailored for causal ablation.
>Add stratified analysis based on scenario difficulty levels. (zZua Q3, Q5kf W3/Q3, Head W2 (c))

We stratify the 14 nuPlan evaluation types into three complexity levels based on dynamic density and interaction difficulty, and report agents' performance within each group. The results show that shortcut reliance remains consistent across all scenario types and becomes more pronounced under higher complexity, confirming that the reasoning–planning disconnect is a fundamental property of the model’s learning behavior. The full breakdown is included in Appendix A.10 (p.20).

**For other concerns, we have provided point-by-point clarifications and additional results:**
- Clarify that our approach is grounded in formal intervention and counterfactual frameworks rather than correlation. (zZua W1)
- Confirm that DriveMind is derived from high-fidelity real-world human driving data and evaluations adhere to established standards. (zZua W2)
- Confirm that token-level co-training or leakage has been ruled out by our design. (zZua W3/Q2)
- Clarify that our CoT is dynamically generated, not redundant, and provide additional experiments demonstrating that the disconnect is not caused by CoT redundancy or insufficient specificity in the training signal. (Q5kf W1/Q1)
- Clarify that all trajectory ground truths originate exclusively from nuPlan (not GPT-4.1) and refine the manuscript accordingly. (Q5kf W2/Q2)
- Demonstrate that no data leakage occurred during CoT generation through prompt design and human verification. (Head W2 (b))
- Provide evidence that our findings are robust across model sizes, datasets, and training recipes. (Head Q1)
- Show that simple solutions like explicit visual grounding prompts or input noise augmentation do not solve the fundamental reasoning-planning disconnect by clarifying existing experiment setups and adding experiments with noisy training. (Head Q2)
- Further validate that models learn textual-prior shortcuts through additional visual-perturbation experiments. (Head Q3)
- Add experiments showing that neither two-stage sequential training nor introducing a visual–CoT alignment reward during RL resolves the disconnect. (xNnf W/Q1/Q3)
- Report the proportion of cases where CoT reasoning diverges from planning outcomes under causal probe intervention. (xNnf Q2)

In summary, we are the first to identify Reasoning–Planning Decoupling in VLM-based driving models, where generated reasoning fails to causally influence planned trajectories. Instead, models rely on shortcut learning from textual priors to generate trajectories; even when the reasoning appears correct, the resulting trajectories can still be incorrect. We provide the community with a new dataset and a quantifiable diagnostic tool to evaluate the causal fidelity.

---

### Meta-Review · Area_Chair_EWAZ · 2026-01-10

**Summary:**

This paper investigates whether the reasoning produced by Vision-Language driving agents truly causes their planning behavior. Authors build DriveMind, a nuPlan-based dataset with plan-aligned Chain-of-Thought reasoning, and perform systematic ablation and perturbation studies on several VLM backbones.

**Reviewer Concerns:**

- The “disconnect” remains correlational; causal claims rely on ablation heuristics rather than formal interventions or counterfactual reasoning frameworks.

- All tests are in simulation (nuPlan) with no real-world validation or human-driving baselines. AC Note: nuPlan is not simulation

- Technical clarity. It is unclear whether every case analyzes the same set of dimensions, or whether the CoT content is truly focused on those features most strongly related to the subsequent driving action.

- Only a small set of illustrative cases, which limits the persuasiveness of the findings.

- Limited contributions.
- Insufficient justification of dataset novelty and quality.
- Limited experimental scope and generalizability.

**Reviewer Scores:**

Although authors respond with very long and detailed feedback, the critical concerns raised by reviewers still remains. In particular, the limitted contribution raised by reviewer Head. It remains unclear that, e.g. (a) whether the observed disconnect is specific due to DriveMind's distributional bias and data quality, or would generalize to other datasets. (b) whether this is an artifact of open-source VLM training or a broader issue.

AC read the paper, rebuttal and review comments. Authors are encouraged to revise the paper in great extent.

---

### Decision · Program_Chairs · 2026-01-26

Reject